# Ocean surface energy balance allows a constraint on the sensitivity of precipitation to global warming

Wei Wang [1,3], T. C. Chakraborty [2,3], Wei Xiao[1] & Xuhui Lee[2 ✉]

Climate models generally predict higher precipitation in a future warmer climate. Whether the precipitation intensification occurred in response to historical warming continues to be a subject of debate. Here, using observations of the ocean surface energy balance as a hydrological constraint, we find that historical warming intensified precipitation at a rate of $0.68 \pm 0.51\%$ K$^{-1}$, which is slightly higher than the multi-model mean calculation for the historical climate ($0.38 \pm 1.18\%$ K$^{-1}$). The reduction in ocean surface albedo associated with melting of sea ice is a positive contributor to the precipitation temperature sensitivity. On the other hand, the observed increase in ocean heat storage weakens the historical precipitation. In this surface energy balance framework, the incident shortwave radiation at the ocean surface and the ocean heat storage exert a dominant control on the precipitation temperature sensitivity, explaining 91% of the inter-model spread and the spread across climate scenarios in the Intergovernmental Panel on Climate Change Fifth Assessment Report.

[1] Yale-NUIST Center on Atmospheric Environment, Nanjing University of Information Science & Technology, Nanjing, China. [2] School of the Environment, Yale University, New Haven, CT, USA. [3] These authors contributed equally: Wei Wang, T. C. Chakraborty. ✉email: xuhui.lee@yale.edu

Quantifying the historical change in global precipitation ($P$) is an important step towards a credible prediction of future $P$ intensification. Currently, climate models disagree on the sign and magnitude of the historical $P$ temperature sensitivity $\Delta P/\Delta T$ [1]. Not helping the matter is the fact that instrumental records of precipitation are land-biased and are too noisy to constrain model calculations[2,3]. Because of the difficulties in detecting changes with observed precipitation[2,3], energy conservation is often used as an alternative to understand the global $P$ trends. The atmosphere will lose more longwave radiation energy to the Earth's surface as its temperature increases due to rising $CO_2$ and as it accumulates water vapor[4,5]. Some of the loss is offset by the water vapor absorption of shortwave radiation, but the majority is balanced by latent heat release accompanied by a greater $P$ [1,6,7]. Although observational constraints regarding the shortwave absorption exist[8], empirical data about the atmospheric longwave loss at the top of the atmosphere (TOA) are still uncertain[9,10]. For this reason, it is not possible to estimate the $P$ change as a residual of the atmospheric energy balance.

Energy balance is also maintained at the Earth's surface, where observational data can provide independent constraints on $P$. A well-known thermodynamic consequence of greenhouse gas-induced warming is the increase of water vapor abundance in the atmosphere. This atmospheric moistening is responsible for about half of the increase in the longwave radiation[4,5] and for almost all the reduction in the clear-sky shortwave radiation[5] or solar dimming at the Earth's surface. Other components of the surface energy balance will also adjust to rising temperatures. Chief among these are the reduction of ocean albedo $a$ associated with melting of the sea ice[11] (Supplementary Fig. 1) and the shift of the ocean Bowen ratio ($\beta$, the ratio of sensible to latent heat flux) towards lower values[12,13]. The historical changes in $a$ and $\beta$ are large, but their effects on $P$ are not known.

Here we develop a surface energy balance constraint on the global hydrological cycle. In this framework, the global precipitation change is partitioned into contributions from observed changes in the energy balance terms of the global ocean surface. We find that historical changes in $a$, $\beta$ and surface longwave radiation intensified global $P$, and changes in surface shortwave radiation and ocean heat storage weakened $P$, with the former slightly outweighing the latter. We then extend the framework to diagnose climate model predictions of future $P$ change, revealing a robust emergent relationship of $\Delta P/\Delta T$ with two key surface energy components.

## Results

**Historical precipitation temperature sensitivity**. We hypothesize that changes in global precipitation $\Delta P$ are driven primarily by changes in ocean evaporation $\Delta E_O$ at the annual and longer time scales. At these time scales, $P$ is balanced by surface evaporation. Accordingly, $\Delta P$ can be expressed as a proportion to $\Delta E_O$ (see "Methods" section). This hypothesis, which is an inference from the proportionality and is implicit in earlier studies of $P$ trends[14–16], is supported by the tight linear relationship between $\Delta P$ and $\Delta E_O$ calculated from both climate model simulations and atmospheric reanalysis (Fig. 1b, linear correlation $R > 0.99$, confidence level $p < 0.001$). The underlying mechanism can be understood with the two interlinked land and ocean components of the global hydrological cycle (Fig. 1a). Higher temperatures trigger high rates of ocean evaporation. Most of the extra water evaporated from the ocean returns to the ocean as precipitation, and some are transported to land by the atmosphere. The extra water coming from the ocean induces stronger land precipitation. Enhanced land precipitation, in turn, raises soil moisture, and consequently land evaporation and runoff also increase. The role of land evaporation change $\Delta E_L$, expressed here as a ratio $\varphi = \Delta E_L/\Delta E_O$ and termed the land modifier (Eq. (2), see "Methods" section), is embedded in the slope of the relationship of $\Delta P$ versus $\Delta E_O$. A positive correlation is found between $\varphi$ and global tree fraction (Supplementary Fig. 2), that is, a lower $\varphi$ associated with the conversion of forests to urban land and

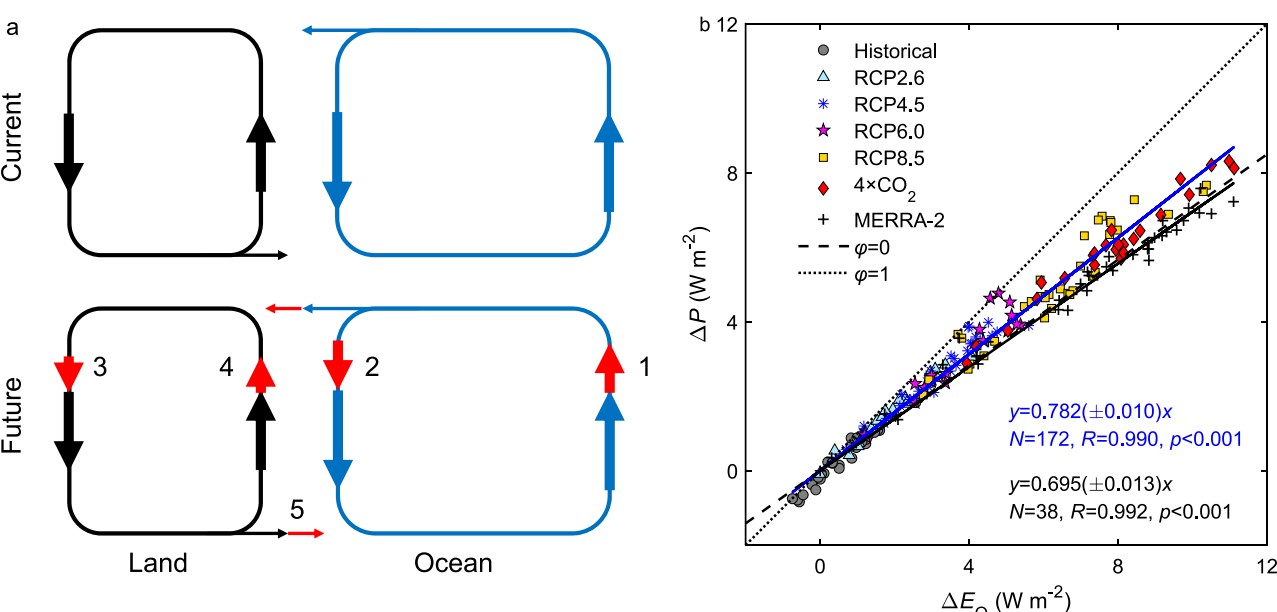

**Fig. 1 Global precipitation change driven by ocean evaporation. a** Two interlinked components of the global hydrological cycle. Arrows 1 to 5 represent changes in ocean evaporation, ocean precipitation, land precipitation, land evaporation, and runoff, respectively. **b**, Relationship between global precipitation change $\Delta P$ and ocean evaporation change $\Delta E_O$ according to CMIP5 model simulations and MERRA-2 reanalysis. The black (reanalysis) and blue solid line (CMIP models) represent linear regression with statistics noted. The dashed and dotted lines represent the land modifier $\varphi$ of 0 and 1. The CMIP results include historical climate (Historical), a low emission scenario (RCP2.6), two medium emission scenarios (RCP4.5 and RCP6.0), a high emission scenario (RCP8.5), and quadrupling of $CO_2$ experiments ($4 \times CO_2$).

cropland in the historical climate and a higher $\varphi$ due to forest regrowth in future climates. This correlation pattern is consistent with studies showing that forests convert precipitation more to evaporation and less to runoff than other land cover types[17], although the increase in stomatal resistance[18] and lengthening of the growing season[19] in a $CO_2$-enriched condition can also affect the land hydrological partitioning. But because land comprises less than 1/3 of the Earth's total surface area, the land modifier effect is diminished substantially in the global precipitation change (Supplementary Table 1).

We combine the ocean surface energy balance (Supplementary Fig. 3) and the proportionality hypothesis to quantify $\Delta P/\Delta T$. Similar to the investigation of evaporation of inland waters[13], here $\Delta E_O$ is partitioned into contributions from changes in $K_\downarrow$ (incoming shortwave radiation), $L_\downarrow$ (incoming longwave radiation) $a$, $\beta$, $L_\uparrow$ (outgoing longwave radiation from the ocean surface), and $G$ (ocean heat storage or heat flux from the ocean surface into the water column). The full equation for $\Delta P/\Delta T$ is

$$\frac{\Delta P}{\Delta T} = s\left\{ -\frac{(R_n - G)}{(1+\beta)^2}\frac{\Delta\beta}{\Delta T} - \frac{K_\downarrow}{1+\beta}\frac{\Delta a}{\Delta T} + \frac{(1-a)}{1+\beta}\frac{\Delta K_\downarrow}{\Delta T} \right.$$
$$\left. + \frac{1}{1+\beta}\frac{\Delta L_\downarrow}{\Delta T} - \frac{1}{1+\beta}\frac{\Delta L_\uparrow}{\Delta T} - \frac{1}{1+\beta}\frac{\Delta G}{\Delta T} \right\} \quad (1)$$

where $R_n = (1-\alpha)K_\downarrow + L_\downarrow - L_\uparrow$ is the ocean surface net radiation, and $s$ is a proportionality coefficient (see "Methods" section). An advantage of performing energy balance analysis over the oceans rather than over the whole globe is that ocean evaporation occurs at the potential rate limited by energy only, whereas land evaporation is confounded by both soil moisture and energy availability and is more difficult to determine from observational data. For this reason, the ocean $\beta$ can be determined with the classic Priestley–Taylor model of potential evaporation. As temperature rises, the vapor pressure at the water surface increases exponentially according to the Clausius–Clapeyron

equation. This results in a faster change in the sea-air vapor pressure gradient than in the temperature gradient, and $\beta$ decreases[16]. Using observational constraints, each of the terms of the energy balance equation is expressed as a function of the global mean temperature, and its temperature sensitivity is given by the slope coefficient of the relationship (Supplementary Table 2). For example, the Bowen ratio temperature sensitivity $\Delta\beta/\Delta T$ of $-0.0083\,\mathrm{K^{-1}}$ is obtained from the Priestley–Taylor model modified on the basis of the Objectively Analyzed Air-sea Flux data set[12]. The surface albedo temperature sensitivity $\Delta a/\Delta T$ is $-0.0065\,\mathrm{K^{-1}}$ according to the measurement made by the Clouds and the Earth's Radiant Energy System (CERES[20]; Supplementary Fig. 1). Scaling the sum of all the component contributions by the proportionality coefficient $s$ for $\Delta P$ versus $\Delta E_O$, we obtain $0.60 \pm 0.44\,\mathrm{W\,m^{-2}\,K^{-1}}$ ($0.68 \pm 0.51\%\,\mathrm{K^{-1}}$; mean $\pm$ 1 S.D) for $\Delta P/\Delta T$ (see "Methods" section). Ocean albedo change contributes positively to the overall sensitivity (Fig. 2a). Melting of the sea ice has long been recognized as positive feedback that amplifies warming. Our result suggests that the same process may also increase the global precipitation temperature sensitivity.

One unresolved question is related to the land modifier $\varphi$. The $\varphi$ value is lower according to the reanalysis data ($-0.05$) than the ensemble model mean for the historical climate (0.15; Supplementary Table 1), despite fixed land use and $CO_2$ concentration in the reanalysis. As mentioned earlier, the estimate of global $P$ change is not sensitive to $\varphi$: use of the climate model mean $\varphi$ would increase $\Delta P/\Delta T$ by only 8%. However, accurate determination of $\varphi$ may be important for the land components of the hydrological cycle.

**Energy balance partitioning of modeled precipitation change.** The same ocean surface energy balance equation is used to diagnose climate model predictions of $\Delta P/\Delta T$ using the CMIP5 (Coupled Model Intercomparison Project 5) simulation results (Fig. 2b–g). Let us first discuss the historical scenario. The

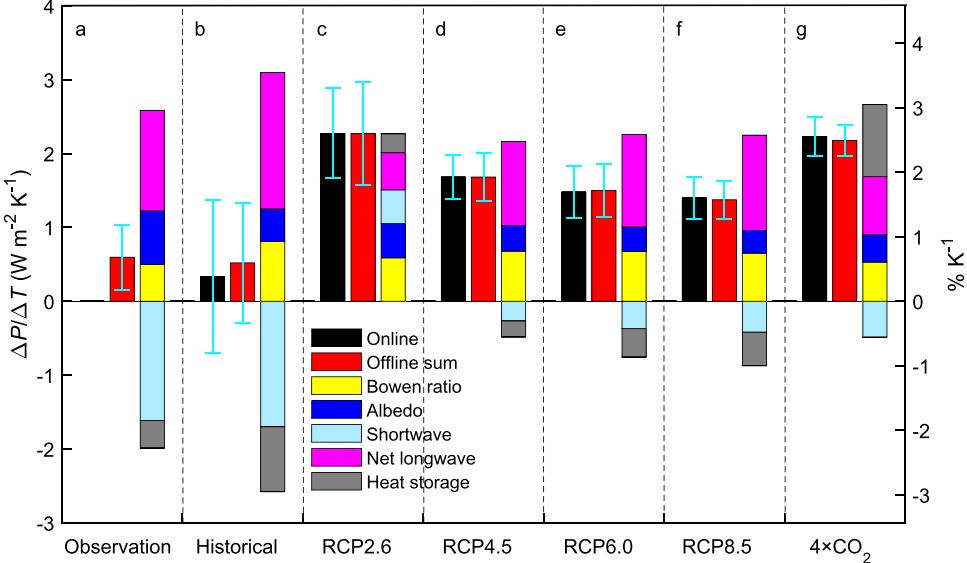

**Fig. 2 Component contributions to global precipitation temperature sensitivity. a** Results from observations of ocean surface energy balance. **b–g** CMIP5 scenarios are labeled at the bottom. According to Eq. (1), global precipitation temperature sensitivity $\Delta P/\Delta T$ is partitioned into temperature sensitivity of changes in Bowen ratio, ocean surface albedo, incoming shortwave radiation, net longwave radiation (incoming longwave radiation minus outgoing longwave radiation), and ocean heat storage. Black: climate model online calculation; red: sum of the five component contributions; yellow: contribution by Bowen ratio change; blue: contribution by surface albedo change; light blue: contribution by a change in surface downward shortwave radiation; magenta: contribution by a change in surface net longwave radiation; gray: contribution by a change in ocean heat storage. Error bars are ± one standard deviation. The magnitude and percentage of $\Delta P/\Delta T$ are given by the left and the right y axis, respectively. Description of scenarios is given in Fig. 1 caption.

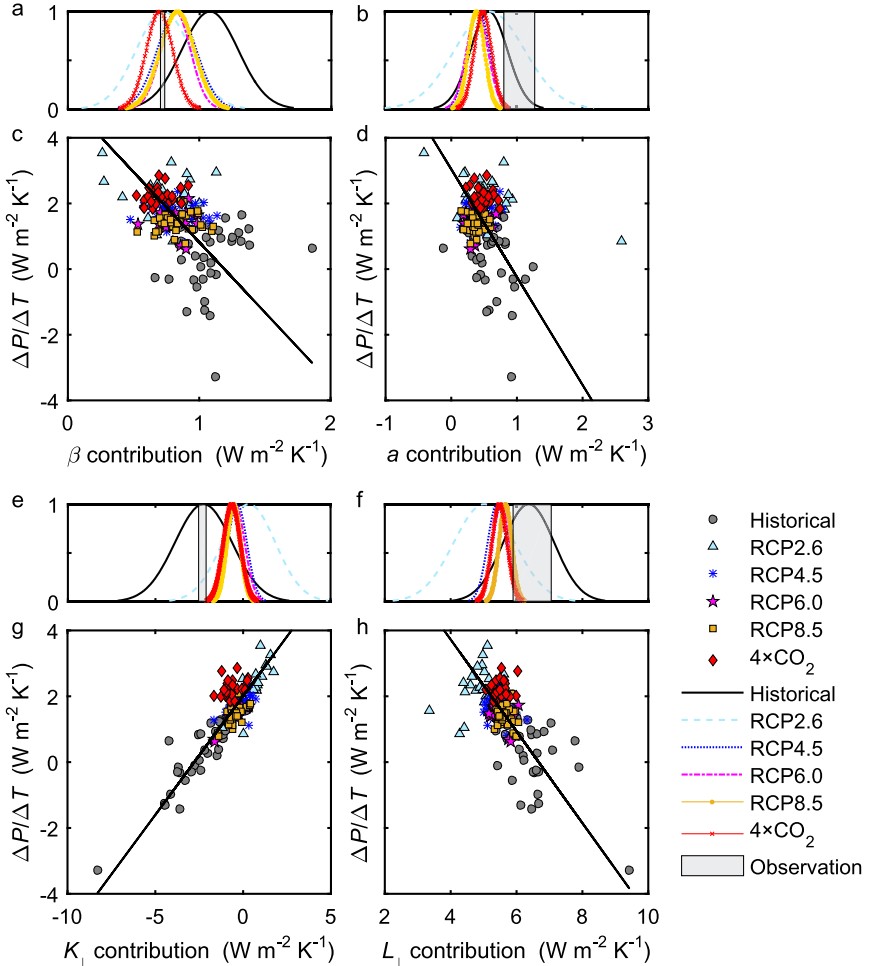

**Fig. 3 Relationship between global precipitation temperature sensitivity and temperature sensitivity of surface energy balance components. c, d, g, h** Relationship between precipitation temperature sensitivity ($\Delta P/\Delta T$) and contribution from change in Bowen ratio $\beta$ $[-(R_n - G)/(1+\beta)^2/(\Delta\beta/\Delta T)]$, surface albedo $a$ $[-K_\downarrow/(1+\beta)(\Delta\alpha/\Delta T)]$, surface downward shortwave radiation $K_\downarrow$ $[(1-\alpha)/(1+\beta)(\Delta K_\downarrow/\Delta T)]$ and surface downward longwave radiation $L_\downarrow$ $[1/(1+\beta)(\Delta L_\downarrow/\Delta T)]$. **a, b, e, f** Gaussian probability distribution of the component contribution. The observational range (mean ± one standard deviation) is denoted by the gray bar. Solid lines are regression fit. Other variables are $R_n$ ocean surface net radiation, $G$ ocean heat storage. Description of model scenarios is given in Fig. 1 caption.

simulation result for the historical climate ($0.33 \pm 1.03$ W m$^{-2}$ K$^{-1}$ or $0.38 \pm 1.18\%$ K$^{-1}$; Fig. 2b) is slightly lower than the observation-constrained $\Delta P/\Delta T$ ($0.60 \pm 0.44$ W m$^{-2}$ K$^{-1}$) and is within the statistical uncertainty of each other (one-sample two-tailed $t$-test $p > 0.12$). Compared to the observational constraint, the ensemble mean albedo contribution to $\Delta P/\Delta T$ is biased low (Fig. 3b). This error is compensated by the high bias of the Bowen ratio contribution (Fig. 3a).

The ensemble model mean $\Delta K_\downarrow/\Delta T$ and $\Delta L_\downarrow/\Delta T$ for the historical scenario are in good agreement with the reanalysis data (Fig. 3e, f), but large inter-model variations are evident. As the climate warms, models that predict more dimming of solar radiation or a more negative $\Delta K_\downarrow/\Delta T$ tend to give stronger incoming longwave radiation or a more positive $\Delta L_\downarrow/\Delta T$ at the ocean surface. This compensatory behavior exists across all the models and all the climate scenarios ($R = -0.830$, $p < 0.001$, number of model simulations $N = 175$; Supplementary Fig. 4). If considered alone, the shortwave contribution to $\Delta P/\Delta T$ shows a large spread of 1.27 W m$^{-2}$ K$^{-1}$ (1 S.D. across all model simulations, Fig. 3g). By combining the contributions of all wavelengths (sum of terms 3 and 4, Eq. (1)), the spread becomes 0.79 W m$^{-2}$ K$^{-1}$, or a 38% reduction in uncertainty. These results support the view that climate models seem to perform better for

composite thermodynamic variables—in this case, the all-wave radiation—than for the individual variables that make up the composites[21].

The tight and negative relationship between $\Delta K_\downarrow/\Delta T$ and $\Delta L_\downarrow/\Delta T$ can be considered as an emergent property of the earth system models. A full cancellation between the surface shortwave and longwave effects requires the slope of $\Delta K_\downarrow/\Delta T$ versus $\Delta L_\downarrow/\Delta T$ to be equal to $-1$. The actual slope is about $-2$, meaning that the dimming of solar radiation $K_\downarrow$ outweighs the strengthening of incoming longwave radiation $L_\downarrow$. The overall result is a reduced precipitation temperature sensitivity if modeled dimming is too strong, and vice versa. Three mechanisms are known to cause a negative relationship between $\Delta K_\downarrow$ and $\Delta L_\downarrow$. Atmospheric moistening at higher temperatures reduces $K_\downarrow$ slightly and increases $L_\downarrow$ by four times as much[5] (Supplementary Fig. 4). The decrease in cloud cover causes a positive $\Delta K_\downarrow$ and a negative $\Delta L_\downarrow$. Aerosols generally reduce $K_\downarrow$ and can enhance $L_\downarrow$ slightly if their size coincides with the wavelengths of the thermal atmospheric window[22]. Evidence points to cloud cover as the dominant cause of this emergent property. This is because clouds have much stronger surface radiative effects than aerosols[23,24]. Furthermore, the strength of the negative cloud shortwave radiative effect is approximately twice that of the positive cloud

longwave radiative effect[20,23], giving a ratio in good agreement with the regression slope of $\Delta K_\downarrow/\Delta T$ versus $\Delta L_\downarrow/\Delta T$. Radiative column calculations suggest that this 2:1 ratio is a radiative property of low clouds[5].

Surface solar radiation change is a large contributor to the spread of $\Delta P/\Delta T$ across scenarios. The RCP2.6 scenario has the highest ensemble mean $\Delta P/\Delta T$ (2.27 W m$^{-2}$ K$^{-1}$; Fig. 2c). In this scenario, the mean $\Delta K_\downarrow/\Delta T$ is actually positive (0.46 W m$^{-2}$ K$^{-1}$), indicating surface brightening as the temperature increases. The lowest ensemble mean $\Delta P/\Delta T$ (0.33 W m$^{-2}$ K$^{-1}$) is found for the historical scenario in which a unit rise in temperature creates the strongest surface dimming ($\Delta K_\downarrow/\Delta T = -1.70$ W m$^{-2}$ K$^{-1}$).

Another contributor to the inter-scenario spread is the change in ocean heat storage. In the $4 \times CO_2$ scenario, a reduction in $P$ occurs at the beginning of the experiment[1,8]. This reduction can be interpreted from the surface energy balance perspective. The sudden quadrupling of atmospheric $CO_2$ causes a large radiation imbalance at the top of the atmosphere and similarly large heat flux into the ocean (multi-model mean $G = 6.52$ W m$^{-2}$ in the first 10 simulation years). This results in less energy available at the surface to support ocean evaporation, and global $P$ declines. At the end of the experiment, $G$ approaches zero as the climate system reestablishes equilibrium, giving rise to a negative $\Delta G/\Delta T$. In the historical scenario and future scenarios with a progressive rise in $CO_2$ (RCP4.6, RCP6.0, and RCP8.5), the role of ocean heat storage is to suppress $\Delta P/\Delta T$ (Fig. 2b, d–f). In contrast, the change in $G$ enhances $\Delta P/\Delta T$ in the $4 \times CO_2$ scenario, contributing to a very high-value $\Delta P/\Delta T$ of $2.23 \pm 0.26$ W m$^{-2}$ K$^{-1}$ (Fig. 2g).

**An emergent property of global precipitation**. The energy balance analysis reveals another emergent property of the earth system models: the incoming solar radiation at the ocean surface $K_\downarrow$ and ocean heat storage $G$ are the two energy terms that exert a dominant control on the global precipitation temperature sensitivity. The combined contribution of these two variables is highly linear with $\Delta P/\Delta T$ and explains 91% of the inter-model and inter-scenario variations in $\Delta P/\Delta T$ (Fig. 4a). Most notably, the combination collapses the $4 \times CO_2$ results onto the same linear relationship that depicts the other transient scenarios. In the $4 \times CO_2$ scenario, the $K_\downarrow$ temperature sensitivity alone is a poor predictor of $\Delta P/\Delta T$ ($R = 0.29$, $p = 0.17$, $N = 25$). However, the correlation becomes significant when both $K_\downarrow$ and $G$ are considered ($R = 0.808$, $p < 0.001$, $N = 25$). In an out-of-sample test, we find that the linear fit line describes reasonably well a CMIP6 model ensemble (Supplementary Fig. 5c).

The emergent relationship in Fig. 4a provides an additional constraint on the historical $\Delta P/\Delta T$. Using the observed $\Delta K_\downarrow/\Delta T$ and $\Delta G/\Delta T$ and the fitting equation in Fig. 4a, we obtain a slightly lower estimate of the historical $\Delta P/\Delta T$ ($0.51 \pm 0.21$ W m$^{-2}$ K$^{-1}$) than the energy balance constraint ($0.60 \pm 0.44$ W m$^{-2}$ K$^{-1}$).

## Discussion
Our results based on the surface energy consideration can be put into the context of atmospheric energy conservation. In the atmosphere, the latent heat released by $P$ change is balanced by changes in shortwave radiation absorption of the atmosphere, in its longwave radiation loss, and in the sensible heat flux from the surface[1,6,7]. To facilitate comparison between these two energy perspectives, we note that the TOA radiation imbalance can be approximated by the ocean heat storage $G$ because $G$ explains ~90% of the imbalance historically[25] and more in the future[26]. Previous findings on the role of atmospheric energy components are broadly consistent with the results presented here. The finding that the absorbed shortwave largely controls inter-model spread

in $\Delta P/\Delta T$ in abrupt $CO_2$ ($4 \times CO_2$ and $2 \times CO_2$) scenarios[8,27] is supported by the $\Delta P/\Delta T$ correlation with $K_\downarrow$ change (Fig. 3g). The role of sensible heat flux in the historical $P$ change[28] can be understood through the correlation with the change in $\beta$ (Fig. 3c). The importance of atmospheric longwave cooling documented for a future transient climate[5] and for the historical climate[29] is manifested in the correlation with changes in $G$ (Fig. 5c) and $L_\downarrow$ (Fig. 3h) because a longwave loss to outer space is a large contributor to the TOA energy imbalance (and hence to $G$). However, when examined individually, these energy components generally lack consistency between within-scenario and inter-scenario variations. For example, the relationship between $L_\downarrow$ change and $\Delta P/\Delta T$ is positive for the $4 \times CO_2$ scenario ($R = 0.25$) but is negative across scenarios (Fig. 3h). In contrast, consistency is achieved if the incoming shortwave at the ocean surface and the ocean heat storage are combined (Fig. 4a). Since $\Delta K_\downarrow$ is approximately equal to the change in atmospheric absorption of shortwave minus the change in the TOA net shortwave radiation, and $\Delta G$ is an approximation of the change in the total net radiation at the TOA, a physical interpretation of the emergent relationship in Fig. 4a is that shortwave absorption (a known source of model spread[8,30]) and longwave loss at the TOA[5] dominate the modeled $P$ change.

The relationship in Fig. 4a reveals additional diagnostic insights regarding the energy constraints on global $P$. It suggests that strong compensatory behaviors exist among thermodynamic processes in the climate system. For example, warming and moistening of the atmosphere give rise to predictable increases in $L_\downarrow$[4,5], but because $L_\downarrow$ and $K_\downarrow$ are tightly coupled (Supplementary Fig. 4), the inclusion of the $L_\downarrow$ contribution does not bring much improvement to the relationship except for rectifying one outlier (Fig. 4c). (The increase in $R^2$ is marginal, from 0.910 in Fig. 4a to 0.912 in Fig. 4c) A numerical perturbation experiment suggests that change in $K_\downarrow$ may also be coupled with change in $\beta$ through changes in low-cloud cover[31]. That $\Delta G/\Delta T$ emerges as a dominant control of $\Delta P/\Delta T$ supports the view that monitoring the ocean heat content could be the best strategy available to constrain future $P$ change[6]. Since global dimming is the other dominant contributor, long-term monitoring of solar radiation at the earth's surface, especially at marine locations, should provide another strong constraint on $P$.

Figure 4a implies a connection between the $P$ temperature sensitivity and the strength of climate feedback. In the abrupt $4 \times CO_2$ scenario, the TOA radiation imbalance decreases and the surface temperature increases over time after the sudden $CO_2$ rise. In the paradigm of radiative forcing versus climate feedback, the slope of the TOA radiation imbalance versus surface air temperature is a measure of the feedback strength[32]. Since $G$ accounts for a great majority of the imbalance, the magnitude of $\Delta G/\Delta T$ obtained from $4 \times CO_2$ simulations can be regarded as a good approximation of the feedback strength. We find that among the CMIP5 ensemble of models, those with a stronger feedback strength tend to give a higher $\Delta P/\Delta T$ in the $4 \times CO_2$ scenario ($R = 0.41$, $p < 0.05$; Fig. 5a). This positive correlation between the hydrological climate sensitivity and the feedback strength is also evident from simulations with one CMIP5 model member (MIROC5) under different states of perturbed ocean evaporation[33]. The feedback strength on its own, however, has a limited ability of explaining inter-model variations for the historical climate and for future transient scenarios (Supplementary Fig. 6).

Opinions are divided as to whether climate models overestimate[33] or underestimate future $\Delta P/\Delta T$[34]. Here, we ranked the CMIP5 models according to how close their historical values of $\Delta K_\downarrow/\Delta T$ and $\Delta G/\Delta T$ are to the observed values and analyzed the results of 1/3 of the models that rank closest to these observations.

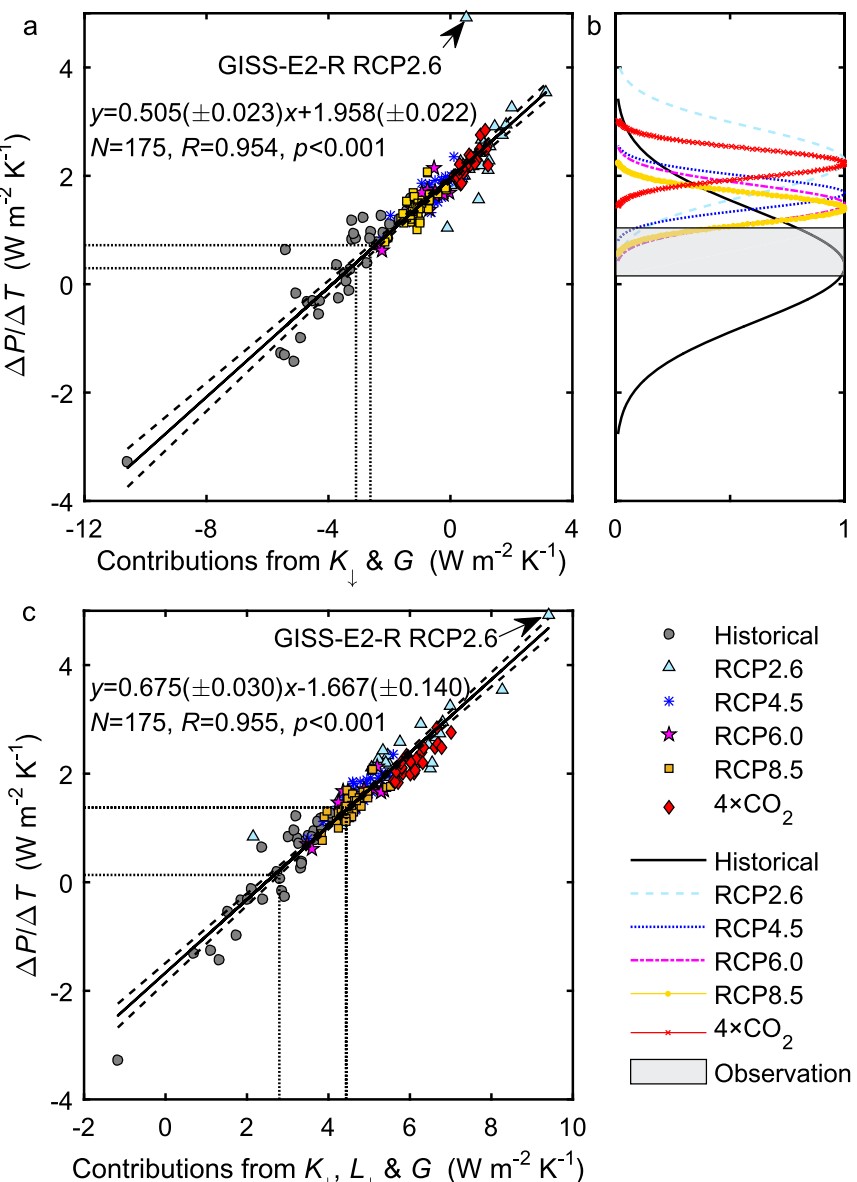

**Fig. 4 Emergent constraint on global precipitation temperature sensitivity. a** Control of surface incoming shortwave radiation ($K_\downarrow$) and ocean heat storage ($G$) on the spread of climate model simulations of precipitation temperature sensitivity ($\Delta P/\Delta T$). The $x$ axis is the combined contribution from changes in $K_\downarrow$ and $G$, as $(1-a)/(1+\beta)(\Delta K_\downarrow/\Delta T)-1/(1+\beta)(\Delta G/\Delta T)$. Solid line represents the best fit with the regression statistics noted. (The outlier is excluded from the regression.) Dashed lines are the 95% confidence bounds. The two vertical and horizontal parallel lines indicate the observational constraint. **b** Gaussian probability distribution of precipitation temperature sensitivity. The gray horizontal bar denotes the observational range (mean ± one standard deviation). **c** Same as **a** except that the surface incoming longwave radiation ($L_\downarrow$) is included as a controlling variable, as $(1-a)/(1+\beta)(\Delta K_\downarrow/\Delta T)+1/(1+\beta)(\Delta L_\downarrow/\Delta T)-1/(1+\beta)(\Delta G/\Delta T)$. Description of model scenarios is given in Fig. 1 caption.

This sub-ensemble mean $\Delta P/\Delta T$ is lower than the whole ensemble mean by 4% (for $4\times CO_2$) to 19% (for RCP2.6). However, this result should be interpreted with caution because the correlations between historical and future contributions from $\Delta K_\downarrow$ and $\Delta G$ across models are statistically insignificant ($R$ in the range $-0.07$ for RCP4.5 to 0.18 for $4\times CO_2$; $p>0.05$). The lack of good correlation suggests that mechanisms that change the surface energy balance may be different between historical and future climates. For example, according to the CMIP5 models, a unit rise in temperature results in less surface solar dimming in the future than in the past (Fig. 3e) despite a similar rate of water vapor buildup of about 7% $K^{-1}$ [6,35–37], in part due to differences in aerosols[38]. It appears that models that are more realistic for the historical climate do not necessarily perform better for future climates.

Our diagnostic analysis (via Eq. (1)) is restricted to the global scale. Even though it has shed light on the manifestation of interactions among energy variables, a mechanistic understanding of the nature of these interactions will require a more granular examination at local and regional levels. Rising temperatures will decrease ocean $\beta$ [12,16]. Since $\beta$ is already very low for mid- to low-latitude ocean regions (about 0.13 between 60° S and 60° N), this thermodynamic response is more important for high-latitude regions where the high $\beta$ (about 0.70 north of 60° N and south of 60° S) allows more room for energy allocation shift from sensible heat to latent heat as evident in historical climate simulations[28]. On the other hand, the high $\beta$ may counteract the increase of radiation energy available for evaporation via a reduction in polar waters. Additionally, changes in $K_\downarrow$ and $a$ at high latitudes are

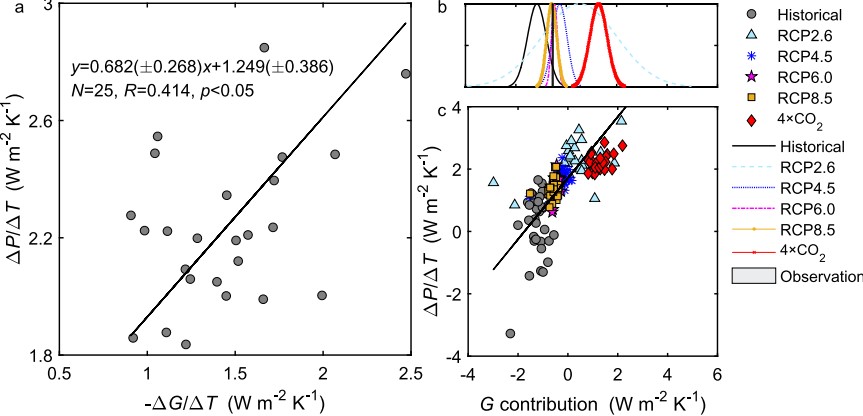

**Fig. 5 Relationship between global precipitation temperature sensitivity and temperature sensitivity of ocean heat storage. a** Dependence of precipitation temperature sensitivity ($\Delta P/\Delta T$) on climate feedback strength for the $4 \times CO_2$ scenario. The climate feedback strength is approximated by a negative value of temperature sensitivity of ocean heat storage ($-\Delta G/\Delta T$) from $4 \times CO_2$ scenario. **b** Gaussian probability distribution of the ocean heat storage ($G$) contribution to $\Delta P/\Delta T$. **c** Relationship between precipitation temperature sensitivity and $G$ contribution [$-1/(1+\beta)(\Delta G/\Delta T)$] across models and scenarios. The observational range (mean ± one standard deviation) is denoted by the gray bar. Solid lines are regression fit. The regression slope in **c** (0.982) is increased to 1.290 if the $4 \times CO_2$ results are excluded. Description of model scenarios is given in Fig. 1 caption.

positively correlated in the CERES data and across the CMIP5 models (Supplementary Fig. 7), consistent with the observation of greater cloud cover during low-ice years near the North Pole[39]. Thus, change in regional $K_\downarrow$ is another process that may counteract the albedo effect on global $P$. At low latitudes, cloud cover change can also influence $P$. Climate models with a higher equilibrium climate sensitivity are shown to have a more positive low-cloud feedback[40–42] and agree better with constraints provided by the cloud behaviors observed in tropical and subtropical oceans than lower sensitivity models[43–45]. Several mechanisms are known to reduce cloud cover in these regions in a future warmer climate, including the breakup of stratocumulus cloud decks[46], aggregation of deep convective clouds[47], and high cloud shrinkage associated with tightening of the ascending branch of the Hadley Circulation[34]. The ocean surface $K_\downarrow$ will increase in response to the reduction in cloud cover, but it is not known if this increase is large enough to offset the dimming caused by rapid water vapor buildup in the tropical and subtropical atmosphere so as to result in a net increase in $P$. Numerical perturbation experiments may be necessary to disentangle the role of these interactive regional processes in the global $P$ response.

## Methods

**Relationship between global precipitation and ocean surface energy balance**. The residence time of water in the atmosphere is about one week[48]. At the annual and longer time scale, mass conservation requires that global precipitation be balanced by surface evaporation. Additionally, three corollaries of the conservation principle apply to the two interlinked components of the hydrological cycle (Fig. 1a): ocean evaporation is equal to the sum of ocean precipitation and water transport to land by the atmosphere, land precipitation is equal to the sum of land evaporation and river runoff to the ocean, and river runoff is equal to atmospheric transport. Using these equality relationships, we obtain

$$\Delta P = \left(f_L \varphi + 1 - f_L\right)\Delta E_O \qquad (2)$$

where $\Delta$ denotes change between two-time intervals, $P$ is mean global precipitation rate and $E_O$ is mean ocean evaporation rate (both expressed in W m$^{-2}$), $f_L$ is land fraction (=0.29), and $\varphi = \Delta E_L/\Delta E_O$ is the land modifier with $E_L$ representing mean land evaporation rate. The temperature sensitivity of global precipitation is given by

$$\frac{\Delta P}{\Delta T} = s\frac{\Delta E_O}{\Delta T} = \left(f_L \varphi + 1 - f_L\right)\frac{\Delta E_O}{\Delta T} \qquad (3)$$

The proportionality coefficient $s$ is weakly dependent on the land modifier. If $\varphi = 1$ or $E_L$ changes at the same rate as $E_O$, $s$ would be at the upper limit of unity. If $E_L$ remains fixed over time or $\varphi = 0$, $s$ would be reduced to 0.71. The actual $s$ values, based on linear regression of climate model results and MERRA-2 reanalysis data, are given in Supplementary Table 1.

The ocean evaporation flux, expressed in the form of latent heat consumption, is in balance with other component fluxes of the ocean surface energy balance (Supplementary Fig. 3). Specifically,

$$R_n = (1 - a)K_\downarrow + L_\downarrow - L_\uparrow = H + E_O + G \qquad (4)$$

or

$$E_O = \frac{(1 - a)K_\downarrow + L_\downarrow - L_\uparrow - G}{1 + \beta} \qquad (5)$$

where $R_n$ is net radiation, $K_\downarrow$ is incoming solar radiation flux at the ocean surface, $a$ is ocean surface albedo, $L_\downarrow$ and $L_\uparrow$ are incoming and outgoing longwave radiation flux, respectively, $G$ is heat flux from the atmosphere to the water column, and $\beta$ is Bowen ratio or the ratio of sensible heat to latent heat flux. In the diagnostic analysis presented above, the energy fluxes in Eq. (5) are area-weighted ocean mean values, $a$ is the ratio of area-weighted mean reflected to incoming solar radiation, and $\beta$ is the ratio of area-weighted mean sensible to latent heat flux. Because the lateral transport of heat via ocean currents is zero at the global scale, $G$ is equivalent to the change in the ocean heat content or heat storage. For these reasons, Eq. (5) is exact at the global scale.

Expressing Eq. (5) in finite-difference form[13] and combining with Eq. (2), we obtain Eq. (1). Equation (1) is the basis for the quantification of historical $\Delta P/\Delta T$ from observational data and for diagnostic analysis of climate model results.

Our analytical framework can be considered an extension of the work by Siler et al.[16] who decomposed future $P$ change with the ocean surface energy balance equation. In their study, the thermodynamic response, or shift of energy allocation from sensible heat to latent heat, consists of a change in the equilibrium Bowen ratio and changes in boundary layer dynamics/relative humidity. It can be shown that their diagnostic equation (their Eq. (16), without the boundary layer term) is identical in form to the terms in the curly brackets of Eq. (1). In this study, the thermodynamic response is determined with the Bowen ratio from the modified Priestley–Taylor model of ocean evaporation[12] and the actual Bowen ratio from sensible heat and latent heat fluxes calculated by climate models. Because the actual Bowen ratio is less sensitive to temperature than the theoretical equilibrium Bowen ratio, this thermodynamic contribution to the global $P$ change is smaller in our assessment. Additionally, we have introduced a land modifier to account for the land evaporation contribution to global $P$.

**Observational constraints on historical precipitation temperature sensitivity**. The temperature sensitivity of the terms in Eq. (1) was determined from their observed relationships with global mean temperature (Supplementary Table 2). (a) *Ocean albedo a*: the annual $a$ is the ratio of the area-weighted annual mean outgoing shortwave radiation to incoming shortwave radiation observed at the global ocean surface by the Clouds and the Earth's Radiation Energy System (CERES Edition 4.1 [20]). The temperature sensitivity of $a$ was calculated as the regression slope of $a$ against the global mean temperature anomaly (the GISS Surface Temperature Analysis; GISTEMP v4) from 2001 to 2018 (Supplementary Fig. 1). The uncertainty (one standard deviation) is approximated by ½ of the 95% confidence bound on the regression slope. (b) *Ocean Bowen ratio ($\beta$)*: according to the modified version of the Priestley–Taylor model on the basis of the Objectively Analyzed Air-sea Flux data set[12], oceanic $\beta$ is inversely proportional to the slope of the saturation vapor pressure versus temperature $T$. The $\beta$ temperature sensitivity was obtained from the derivative of this function with respect to $T$ and evaluated at

the observed global mean temperature. It was computed for each year, and the spread corresponds to one standard deviation of the interannual variability. (c) *Incoming solar radiation* $K_\downarrow$: We used four atmospheric reanalysis datasets (NOAA-CIRES, NCEP-NCAR, JRA-55, and ERA-5; Supplementary Table 3) to determine $\Delta K_\downarrow/\Delta T$. We first established a linear relationship between the annual area-weighted $K_\downarrow$ over the ocean grids and the annual mean global 2-m air temperature from the same reanalysis. The slope of the relationship was taken as $\Delta K_\downarrow/\Delta T$. We then adjusted the slope value slightly by the percent bias of the reanalyzed $K_\downarrow$ in reference to the CERES $K_\downarrow$ for the common period (2005–2013). The $\Delta K_\downarrow/\Delta T$ values given in Supplementary Table 2 are the mean and standard deviation of the four reanalysis datasets. MERRA-2 is excluded because its $\Delta K_\downarrow/\Delta T$ ($-9.16$ W m$^{-2}$ K$^{-1}$) is too negative. Changes in $K_\downarrow$ due to $CO_2$ absorption are negligible[49]. (d) *Incoming longwave radiation* $L_\downarrow$: The $L_\downarrow$ temperature sensitivity was also obtained from the reanalysis products (MERRA-2 included) and with calibration against the CERES $L_\downarrow$ value. In the reanalysis, atmospheric $CO_2$ concentration is fixed over time. According to a column radiative calculation, doubling of atmospheric $CO_2$ increases $L_\downarrow$ by 1.38 W m$^{-2}$ [49]. Assuming $\Delta T = 3.2 \pm 1.3$ K at $CO_2$ doubling[50], this gives an additional sensitivity of $0.43 \pm 0.11$ W m$^{-2}$ K$^{-1}$. The $\Delta L_\downarrow/\Delta T$ value in Supplementary Table 2 has included this $CO_2$ effect. (e) *Outgoing longwave radiation* $L_\uparrow$: The $L_\uparrow$ temperature sensitivity was given by the derivative of the Stefan–Boltzmann law. The calculation was done annually using the observed global mean temperature. Its uncertainty corresponds to one standard deviation of the interannual variability. (f) *Ocean heat storage G*: $\Delta G/\Delta T$ was obtained by a quadratic fit of the ocean heat content[51] (OHC, in J) against time ($t$) as OHC $=\alpha_0 + \alpha_1 t + \alpha_2 t^2$. The first derivative of OHC with respect to $t$ gives heat storage change or total heat flux (in W) into the water column, and the second derivative (or $a_2$) represents the time rate of change of this total heat flux. Dividing the coefficient of the quadratic term $\alpha_2$ by the ocean area gives the time rate of change of the heat flux into the water column per unit surface area (in W m$^{-2}$ s$^{-1}$), and by multiplying this rate by the length of the observational period (1955–2017), we obtained $\Delta G$. The uncertainty on $\Delta G$ was estimated as ½ of the 95% confidence bound on $\alpha_2$. We then estimated $\Delta G/\Delta T$ by dividing $\Delta G$ with the temperature change $\Delta T$ of 0.774 K observed over the same period according to GISTEMP v4.

We determined the historical $\Delta P/\Delta T$ with Eq. (1) using the sensitivity and the mean ocean surface flux values given in Supplementary Table 2 and the proportionality coefficient $s$ determined from MERRA-2. In Fig. 1b, the MERRA-2 $\Delta P$ and $\Delta E_O$ are deviations of the annual mean $P$ and $E_O$ of a given year from their counterparts in 1985, the year with the lowest $P$. MERRA-2 was chosen for fixing $s$ because this reanalysis model maintains atmospheric moisture conservation[52], which is a prerequisite for Eq. (1). In the other reanalysis products, the global total precipitation can deviate from the global total evaporation by as much as 14%.

The uncertainty of $\Delta P/\Delta T$ was determined with a Monte Carlo method involving 1,000,000 ensemble members. For each member, each term on the right-hand side of Eq. (1) was the sum of its mean value (Supplementary Tables 1 and 2) and a random error produced by a random number generator. This error was assumed to vary independently from other terms and according to a normal distribution with the standard deviation given in Supplementary Table 1 or 2. The uncertainty of $\Delta P/\Delta T$ was calculated as one standard deviation of the ensemble after the top and bottom 0.5% of outliers were excluded.

The linear correlations of $L_\downarrow$ and $K_\downarrow$ with $T$ are highly significant ($p < 0.0001$) for all the five reanalysis products. Although strictly reanalysis $L_\downarrow$ or $K_\downarrow$ are model-derived, the reanalysis model calculations have ingested observational data on temperature, humidity, and cloud, and therefore provide realistic surface radiation fields[53]. Additionally, reanalysis models deploy more accurate codes for shortwave radiation transfer than earth system models[8]. We note that of the five reanalysis products, only NOAA-CIRES extends back in time to coincide roughly with the period of CMIP5 historical simulations. If we use the NOAA-CIRES $\Delta K_\downarrow/\Delta T$ ($-2.36$ W m$^{-2}$ K$^{-1}$) and $\Delta L_\downarrow/\Delta T$ (6.71 W m$^{-2}$ K$^{-1}$, adjusted for the $CO_2$ effect), the historical $\Delta P/\Delta T$ will decrease slightly to 0.37 W m$^{-2}$ K$^{-1}$.

**Diagnostic analysis of CMIP5 model results**. Equation (1) was used to separate the $P$ temperature sensitivity in CMIP models into component contributions. The results presented in the main text were based on one ensemble member (r1i1p1) from six CMIP5 experiments (Historical, RCP2.6, RCP4.5, RCP6.0, RCP8.5, and $4 \times CO_2$) with a total of 176 model simulations (Supplementary Table 4). To perform an out-of-sample test, we also analyzed one ensemble member (r1i1p1f1) of one CMIP6 experiment (ssp585). Scenario ssp585 is an energy and resource-intensive socioeconomic scenario for the 21st century resulting in a similar 2100 radiative forcing (8.5 W m$^{-2}$) as its CMIP5 predecessor RCP8.5. Here, $\Delta$ denotes the difference in a variable between the mean of the last 10-years and that of the first 10-years of the model simulation. In the $4 \times CO_2$ scenario, $\Delta P/\Delta T$ is equivalent to the hydrological sensitivity parameter defined by Fläschner et al.[1] and represents the slow response of $P$ to warming (Supplementary Fig. 8). The $P$ response to warming analyzed by Siler et al.[16] is similar to the apparent hydrological sensitivity given by Fläschner et al.[1]. Fast $P$ adjustment, taken as the $y$ intercept of the $P$ versus temperature regression for the $4 \times CO_2$ simulation in reference to piControl[1], and fast $P$ response (the $P$ difference between sstClim and sstClim4 $\times CO_2$ simulations[16]) are not considered in this study.

The $G$ term is the net heat flux entering the liquid water column plus a small amount of energy consumption due to ice melt at high latitudes. In the above diagnostic analysis, $G$ was calculated as the residual of the ocean surface energy balance equation $G = R_n - H - E_o$ (Supplementary Fig. 3). The surface net radiation ($R_n$) and the ocean sensible ($H$) and latent heat flux ($E_o$) were obtained from the atmospheric data set archived for each model simulation. This residual calculation ensures that energy is conserved in our diagnostic analysis.

Both online and offline $\Delta P/\Delta T$ values are presented in Fig. 2 and Supplementary Fig. 5. The online value is calculated from the modeled $\Delta P$ and $\Delta T$. The offline value is the sum of the component contributions according to Eq. (1). The ensemble mean online and offline values are in excellent agreement for all the scenarios except the historical climate. The offline historical mean is $0.52 \pm 0.81$ W m$^{-2}$ K$^{-1}$, whereas the online historical mean is slightly lower, at $0.33 \pm 1.03$ W m$^{-2}$ K$^{-1}$, but the difference is not statistically significant (two-tailed $t$-test $p = 0.42$). The consistency between the online and offline calculations indicates that Eq. (1) is a robust decomposition procedure and that errors in the global $\Delta P/\Delta T$ arising from spatial averaging of input variables may be small. The offline $\Delta P/\Delta T$ from the surface energy balance (Eq. (1)) for the $4 \times CO_2$ scenario ($2.18 \pm 0.21$ W m$^{-2}$ K$^{-1}$; Fig. 2g) agrees well with the $\Delta P/\Delta T$ diagnosed from the atmospheric energy balance (2.03 W m$^{-2}$ K$^{-1}$ [1]), offering further support for the surface diagnostic method.

To further investigate possible errors due to spatial averaging, we performed a regional diagnostic analysis using CMIP historical simulations. At regional and local scales, the heat flux from the atmosphere to the water column $G$ consists of lateral heat transport by ocean currents and time change in local ocean heat content[54]. Regional analysis is not feasible with observational data because no gridded data exist on the transport term, but it can be done with CMIP modeling outputs as the modeled $G$ includes both lateral heat transport and local heat storage. In this analysis, the ocean grids were divided into two groups: those belonging to mid- and low-latitude regions (between 60° S and 60° N) and those belonging to high-latitude regions (north of 60° N and south of 60° S). The decomposition was performed for each group and the result was weighted by the area fraction of each to obtain a global mean value (Supplementary Fig. 9). The albedo contribution from the two-region analysis is smaller than that from the global analysis. The reduction in the albedo component is offset by less negative contributions from changes in shortwave radiation and in ocean heat storage. The total $\Delta P/\Delta T$ is unaffected, as $\Delta P/\Delta T$ from the two-region analysis ($0.51 \pm 0.76$ W m$^{-2}$ K$^{-1}$; Supplementary Fig. 9b) is nearly identical to that from the global analysis ($0.52 \pm 0.81$ W m$^{-2}$ K$^{-1}$; Supplementary Fig. 9a).

## Data availability
The data used in this study are available at the following public websites: CMIP5 data at https://esgf-node.llnl.gov/search/cmip5/, CMIP6 data at https://esgf-node.llnl.gov/search/cmip6/, NOAA-CIRES and NCEP-NCAR reanalysis at https://psl.noaa.gov/, JRA-55 reanalysis at https://rda.ucar.edu/, ERA-5 reanalysis at https://cds.climate.copernicus.eu/, MERRA-2 reanalysis at https://esgf-node.llnl.gov/search/create-ip/, CERES observations (Edition 4.1) at https://ceres.larc.nasa.gov/data/, global ocean heat content observations at http://159.226.119.60/cheng/, and GISS Surface Temperature Analysis (GISTEMP v4) at https://data.giss.nasa.gov/gistemp/. The data used to produce Figs. 1–5 and Supplementary Figures are available from the authors upon request.

## Code availability
The codes used for data analysis are available from the authors upon reasonable request.

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

## Acknowledgements

This research was supported by the National Key R&D Program of China (Grant 2019YFA0607202 to W.W. and W.X.) and a Yale University Graduate Fellowship (to T.C.). We thank Benjamin Fildier and two anonymous reviewers whose constructive comments have improved this submission.

## Author contributions

X.L. designed the research, developed the conceptual framework, and wrote the manuscript. W.W. analyzed the CMIP modeling data and prepared the online supplement. T.C. analyzed the atmospheric reanalysis data. W.X. contributed ideas to data analysis.

## Competing interests

The authors declare no competing interests.

## Additional information

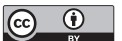

