## [Peer Review File · Nature Communications]

Reviewer comments, first round:

Reviewer #1 (Remarks to the Author):

Overall assessment:

This study employs a rather unique approach (a breakdown of the ocean surface energy budget) to gather physical insight on the global precipitation response to warming and to provide an independent observations-based constraint on this response over the recent past. The authors take the approach a step further to place a constraint on the future projected change of global precipitation in models, concluding that models may be overestimating the precipitation sensitivity. Overall, the study is fairly solid (i.e., adequate use of data and simulations), and, to my knowledge, the approach (i.e., employing the ocean surface energy budget) is novel. The study provides some interesting insights, such as the importance of ocean surface albedo for hydrologic sensitivity, and offers evidence that climate model simulations of historical precipitation change are realistic. There are, however, several issues with aspects of the study. The main issues relate to the practical/physical value of the constraint on future precipitation change, physical interpretation of the constraint on historical precipitation sensitivity in models, and the presentation of methods – more details are provided below. If these issues can be adequately addressed by the authors, I feel the paper may be suitable for publication in Nature Communications and has the potential to be a meaningful contribution to the field.

General comments:

1) Constraint on future DP/DT

I am concerned about the practical value and physical significance of the constraint on future precipitation change (DP/DT) that is presented in the Summary section and Table 1. One important aspect that is overlooked is whether the simulated changes during the historical period are statistically and physically linked to projected changes for the future scenarios in models generally. In other words, is there a robust inter-model relationship between the historical and future sensitivities (of K and G terms, and/or their sum) that would support using the observed historical values to constrain the future projected DP/DT? A statistically significant positive correlation between the historical and future scenario(s) sensitivities across models would offer such support. Have the authors looked at such correlations? Based on the results presented in the paper, it almost seems as if the more realistic models (i.e., the sub-ensemble) tend to have higher historical DP/DT values but lower future DP/DT (comparing Fig. 4a and Table 1, for example) – potentially suggesting a negative cross-model correlation between the historical and future scenario K/G term sensitivities to warming. If such negative relationships indeed exist, how can they be explained physically? If no relationships exist, it could suggest that historical changes are fundamentally distinct from future changes (e.g., different forcing agents and/or physical mechanisms) or that the sample size is inadequate, both of which would substantially limit the value of constraining the future P change with the method presented here.

Related to the above is the somewhat unsatisfying spread among the sub-ensemble, with a STD that is often as large or larger than that of the full ensemble (Table 1). In fact, the range of the sub-ensemble DP/DT values (mean +/- 1 STD) encompasses the mean of the full ensemble for all scenarios, suggesting the constraint on future DP/DT is not statistically meaningful.

I recommend the authors further explore the issues raised above, particularly the inter-model relationships between historical and future scenario sensitivities. If such relationships cannot support the constraint procedure currently presented in the paper, then I suggest the authors go

in a different direction regarding this aspect of the paper. This could mean searching for another way to place a constraint on the future DP/DT (such as focusing on just the surface shortwave or G terms separately, either of which could possibly exhibit satisfying inter-model relationships between historical and future scenarios). Alternatively, removing the quantitative constraint for the future scenarios entirely and replacing it with a discussion (which highlights the challenges of such an approach) may be another option.

2) Interpretation/justification for K-G constraint

The physical interpretation of the sum of the surface shortwave (K) and G terms of Eq. 4, used to constrain the historical P sensitivity in models (Fig. 4), is not adequately addressed in the paper. For example, it is not entirely clear why this combination of terms evidently yields the best correlation with DP/DT across models. Several related questions arise: How strong is the relationship between DP/DT and the sum of surface longwave and shortwave components (the latter which are anti-correlated as described in the manuscript)? Is G itself correlated with the surface longwave component? Is an observational constraint that is based on G and surface SW fluxes (as presented here) necessarily more accurate/reliable than one that includes longwave fluxes?

Regarding the physical interpretation, one could argue that G effectively represents the top-of-atmosphere radiative imbalance (i.e., if one reconciles the surface and atmospheric energy budgets mathematically). Indeed, the authors hint at this when describing characteristics of the 4xCO₂ scenario on page 8. With this interpretation, the sum of the K and G terms (i.e., K term minus G term) would equate to, I think, a sum including the atmospheric shortwave absorption and outgoing top-of-atmosphere longwave radiation – essentially the atmospheric energy budget excluding the surface sensible heat and surface longwave components. Is this why the correlation in Fig. 4 is so high, because it is dominated by shortwave absorption (a known source of model spread)? Can one then obtain another independent observational constraint using measurements of the shortwave radiative energy budget and outgoing longwave radiation instead?

In summary, the energy-based constraint on historical DP/DT in models (Fig. 4) is presented in a way that is more mathematical than physical and that raises a number of unresolved questions (some examples of those questions given above). I think this component of the paper would benefit from a more thorough discussion that includes more physical explanation, justification of the approach, and discussion of potential advantages of this approach versus other similar or equivalent approaches. This may potentially require additional calculations.

3) Methods not adequately described

A key theme of this paper is using observations to place constraints on DP/DT. Yet, many details of the methods for obtaining observational estimates and their uncertainties are glossed over. Additionally, some calculations from the models are not adequately described. The lack of information would make reproducing (or expanding upon) the results difficult if not impossible for the research community, and also makes it hard to judge potential shortcomings of the approach. Specific examples are given below, and should be addressed by adding more details to the text and/or tables in the appropriate places:

- Table S1: It is not clear how the model s values are obtained. Is it done the same way as for MERRA-2 (i.e., a regression across years) or a different way? A follow-up question is: exactly how are the standard deviations computed, in either case?
- How is G computed in models?
- More information about the model scenarios is warranted. For example, what does ssp585 mean and how does it compare qualitatively with the CMIP5 scenarios? Which years specifically are used from each scenario? Perhaps add this information to an existing table (S1?) or create a new table.
- Regarding methods for observational estimates (starting L292): It appears the estimates are in part taken from references and in part derived by the authors, but the degree of each is not

always clear. For example, is Beta purely taken from ref. 12? How much of the G calculation was already done by ref. 37 and how much was done by the authors? More details about the procedures for deriving Beta and G are also warranted for reproducibility, especially considering the importance of G for constraining DP/DT in models.

- Also regarding observational estimates: Which specific years are used from each of the reanalysis datasets? Is the GISS temperature used for all terms?

- Also regarding observational estimates: The quantification of uncertainty is hastily and inadequately described. For example, what constitutes the 1,000,000 ensemble members? More explanation and details are warranted.

Other specific comments:

L29: "empirical data about the atmospheric longwave loss are still highly uncertain" - Two recent studies that are relevant to this statement, which are not currently cited in this manuscript, are Su et al. 2017 (<https://www.nature.com/articles/ncomms15771>) and Watanabe et al. 2018 (<https://www.nature.com/articles/s41558-018-0272-0>): independent observational constraints on DP/DT that are based on longwave measurements. The authors should consider incorporating these references into the discussion here.

L66: A reference to Methods/equation 1 would be helpful.

L78: Add reference to Methods/equation 4.

L94: It's rather important to add a reference to the Methods here.

L99-100: "The ϕ value is lower according to the reanalysis data (-0.05) than the ensemble model mean for the historical climate (0.15; Table S1)" - If s for the models is estimated as a regression across models (rather than regression across years, as for MERRA-2), could this be why the phi values are different? If so, it would be informative to compute phi in models using a regression across years as well, to not only address this question but also potentially provide another line of evidence for a strong relationship (R) between global P and oceanic E.

L137-146: What role does water vapor play in this relationship, if any? I think this is worth including in the discussion here, as water vapor appears a few other times throughout the paper in other contexts.

L294-299: This information regarding the computation of $\Delta\alpha/DT$ should be included, to a reduced extent, in the Fig. S1 caption, since the reader will likely look at Fig. S1 before reading this part of the Methods.

L312-313: "Assuming $\Delta T = 3$ K at CO2 doubling, this gives an additional sensitivity of 0.46 W m⁻² K⁻¹." As demonstrated in ref. 31, this value of warming at CO2 doubling is not entirely known and is unconstrained. I therefore recommend incorporating additional uncertainty into this component.

Acknowledgements/ data statement: I do not see an acknowledgements section stating where the data used for the paper can be accessed. Such a section is important for reproducibility.

Reviewer #2 (Remarks to the Author):

I would like to acknowledge the authors' work, the quality of the text and the effort to provide a different perspective on constraining global rainfall with a different argument. I recommend for publication with minor revisions. Some comments are mostly semantic in order to improve clarity, and some include a few questions on the results and the sources of uncertainty. A more thorough discussion on physical link between this new constraint and the standard one based on

atmospheric energy balance could also strengthen the manuscript.

- L13: please clarify the term "driver"; by "drivers of dP/dT " do you mean "source of intermodel spread in dP/dT "? If instead you mean "driver of change in P ", then ocean albedo mentioned just above also seems to be an important driver. In general throughout the paper, it would be helpful to make a distinction between what is responsible for the change dP/dT (the drivers) and what is responsible for the spread across models or additional sources of uncertainty missing from models. I agree that in the spirit of Allen and Ingram (2002), both are interchangeable, but it seems like it is not the case in your study.

- L15: clarify why the high bias of "4% to 19%" does not match the original discrepancy in the dP/dT found to be "3-4 times smaller than the rates projected" (L9). Just by reading the abstract, it is not clear whether these two statements are supposed to match; it seems so, as it seems that L9 was a motivating question. Is dP/dT still 3-4 times smaller in the historical runs than in the future transient runs also when looking at the sub-ensemble (L198-199)? If so, what explains the remaining difference between the historical dP/dT and future dP/dT ?

- L26: "the atmosphere will lose more LW radiation [...] as it accumulates CO_2 " is correct, but it seems necessary to mention that it is also largely due to water vapor effects in the LW: the next sentence highlights the opposing effect water vapor in the SW, and as currently stated the paragraph implicitly suggests that CO_2 and H_2O oppose each other, which is a little bit too simplistic.

- L29-30: that is a good argument;

1. Pendergrass and Hartmann (2014) also argue that P is more strongly constrained by the atmospheric energy budget than the surface energy budget because of the equilibration time scale of the ocean being larger. Not sure this point is totally relevant, but maybe you could discuss this time scale question here, as well as how it affects your results? For instance in the $4\times CO_2$ run, which shows an energy imbalance in the ocean.

2. "it is not possible to estimate the P change as a residual of the atmospheric energy balance"; true, but from your results, it is not possible to estimate it from a residual in the ocean energy balance either, unless we have reliable estimates of ocean heat uptake, correct? Maybe the importance of G (and whether or not we have confidence on its estimates) could be emphasized in the conclusion a bit more?

- L45 and L63-73: the contributions come from the ocean energy balance, and the estimation of the uncertainty as well. Could you estimate the role of land use change, land drying/wild fires and changes in vegetation/desertification on the uncertainty in dP/dT for future climates? How would ϕ affect the uncertainty range provided L188? Would we get an error range that is larger than the one obtained from the atmospheric budget argument?

- L84-85: I am not familiar with the Priestley-Taylor model. Could you state the physical reason for the change in Bowen ratio with warming? Why would it be more efficient to have a larger fraction of surface cooling by increasing the contribution from evaporation than that of dry turbulent fluxes?

1

- L95-97: please clarify whether these two paths (effect of sea-ice melting on warming and on precipitation) are actually physically distinct and what makes them distinct. In both cases precipitation increases because of an increased latent heat flux from the surface (either via a T increase or via an albedo decrease which allows for more surface absorption). The question is actually : are you referring to P itself or dP/dT ? Is dP/dT physically independent of T ? It seems that by "melting of sea ice amplifies warming" you refer to the direct heating effect of downwelling radiation at the surface, and by "melting of sea ice also intensifies precipitation" you are referring to some additional/marginally increasing fraction of the absorbed radiation that is converted into latent heat, which is not P but dP/dT . Am I correct?

- L146: this would be interesting to discuss the role of low clouds on the uncertainty itself, potentially in the conclusion where you also mention low clouds. You mention they are the key to explain the relationship between dK and dL , so they are key to argue that dK is an important driver of dP . But in parallel, we know the dynamics of low clouds can change with warming.

- L287: it would be helpful to see equation (4) in the main text (or some simplified form of it).

- L345-346: do the atmospheric constraint and the ocean constraint give the same estimate for the run $4\times CO_2$ (see comment made for L29-30)?

--- References

Myles R. Allen and William J. Ingram. Constraints on future changes in climate and the hydrologic cycle. *Nature*, 419(6903), 2002. ISSN 00280836. doi: 10.1038/nature01092.
Angeline G Pendergrass and Dennis L Hartmann. The Atmospheric Energy Constraint on Global-Mean Precipitation Change. *Journal of Climate*, 27(2):757–768, jan 2014. ISSN 0894-8755. doi: 10.1175/JCLI-D-13-00163.1. URL <http://journals.ametsoc.org/doi/abs/10.1175/JCLI-D-13-00163.1>.

Reviewer #3 (Remarks to the Author):

In this paper, the authors use the surface energy budget to decompose the change in global precipitation with global warming into contributions from changes in albedo, the Bowen ratio, net surface radiation, and ocean heat storage. Within this framework, they find a significant role for changes in ice albedo, contradicting earlier studies. They find that shortwave radiation and ocean heat storage account for a large fraction of the intermodal spread. They then propose an emergent constraint on global hydrologic sensitivity to climate change based on observed changes in shortwave radiation and ocean heat storage.

While there are aspects of the study that I find interesting and valuable, I think it has a few serious problems that should prevent it from being published.

1. I'm skeptical that the decomposition in Eq. 4 can be applied in the global mean in a way that's physically meaningful. The reason is that the Bowen ratio is generally smaller where temperatures are warm. Thus, the response of evaporation to changes in radiation or ocean heat storage is quite sensitive to where/when those changes occur: If they occur in regions or seasons in which the Bowen ratio is large (e.g., at high latitudes), their contribution to evaporation change will be quite small.

I suspect this explains why the authors find a large role for changes in ice albedo, in contrast to previous studies. In Eq. 4, the efficacy of albedo change is controlled by $1/(1+\beta)$. Since albedo changes are concentrated at high latitudes, it would be appropriate to use the value of β at these latitudes. By instead using the (smaller) global-mean value of β , the contribution from the change in albedo is likely exaggerated. This would also explain the compensation between changes in albedo and changes in the Bowen ratio, as noted in lines 116-117.

2. The authors lump together the direct effects of CO₂ and warming, but I think it's important to think of these as separate. By itself, an increase in CO₂ causes a decrease in evaporation through an increase in ocean heat storage. This is sometimes called the "fast" response to CO₂, since it is not mediated by temperature change. The "slow", or temperature-mediated response represents the direct impact of warming. The way the authors define the changes in each variable (last 10 years minus first 10 years) doesn't distinguish between these effects, so it's hard to understand what's going on.

A good example of why this is a problem is evident in Fig. 3h, which shows the change in P vs. the change in longwave radiation. There's a clear difference between the inter-ensemble regression slope, which is negative, and the regression slope within a given ensemble, which is harder to discern, but appears to be slightly positive. I'm guessing the overall negative slope is mostly driven by differences in forcing, which don't exist within a given ensemble.

Similarly, the change in ocean heat storage is quite sensitive both to CO₂ forcing and to atmospheric warming, since it is roughly equal to the net radiation imbalance at the top of the atmosphere. We don't know what's going on physically when these effects are lumped together.

3. The authors don't sufficiently engage with previous work. For example, Siler et al. (2019) perform a similar decomposition derived from the Penman equation. The authors should address how their work differs from and builds on this work and other related decompositions based on the atmospheric and surface energy budgets.

Line-by-line comments

Abstract: "find that historical warming intensified P at a rate of 0.39 ± 0.40 %/K, which is ~3-4 times smaller than the rates projected for future transient climates": The authors seem to be implying that future projections are inconsistent with observations, but that's not necessarily true. The sensitivity should be larger the closer the climate is to equilibrium.

25. "The atmosphere will lose more longwave radiation energy to the outer space ... as it accumulates CO₂." This is wrong: increasing CO₂ causes a reduction in longwave emissions to space (hence the greenhouse effect).

95. See above; the global mean decomposition doesn't actually tell us anything about how a decrease in albedo affects global evaporation. If the Bowen ratio is large at high latitudes, a change in albedo would mainly affect the sensible heat flux according to Eq. 4.

112-117: see #1 above

134- : see #2 above. It's hard to interpret this slope without knowing the independent contributions from CO₂ and temperature.

166 - 169: The authors seem misguided here. In prescribed-SST simulations, there is no thermal coupling between the atmosphere and the ocean, so the ocean heat storage term can be quite large, both locally and globally. If SSTs are prescribed to be warmer than their equilibrium value given the prescribed forcing, then there is a net transfer of heat from the ocean to the atmosphere (i.e., ocean heat storage is negative). This will result in a larger increase in evaporation than would realistically occur in response to greenhouse warming.

reference: <https://doi.org/10.1007/s00382-018-4359-0>

Response to reviews of NCOMMS-20-19680 “Constraining the sensitivity of global precipitation to warming with ocean surface energy balance”

(Note: The review comments are in bold font, our responses are in regular font, and changes made to the manuscript are in quotations and highlighted in blue.)

Summary

Thank you very much for your constructive comments. They have significantly improved our manuscript. Before providing a point-by-point response, we wish to summarize the key changes made in this revision:

- 1) Discussion: we have added a discussion section where we put our results in the context of atmospheric energy balance, interpret the physical implications of the emergent relationship, and explain the limitation of our global diagnostic approach.
- 2) Relationship to the study by Siler et al. (2019): Our analytical framework can be considered an extension of Siler et al. (2019). We have expanded the explanation on the connection of our work to this paper.
- 3) Temperature sensitivity: We have clarified the meaning of $\Delta P/\Delta T$, pointing out that it captures the slow response of global P to warming but omits the fast P adjustment.
- 4) Figures and Tables: We have added five new figures (Figures 4c, 5, S6, S7 & S8) and one new Table (Table S4).
- 5) Computation error: We made a small error when computing the longwave radiation temperature sensitivity from MERRA-2. In addition, the sensitivities of Bowen ratio and outgoing longwave radiation were based on model-calculated temperature. In this revision, they have been calculated with the observed temperature. The updated estimate of the historical $\Delta P/\Delta T$ from the surface energy balance is in better agreement with the value constrained by the emergent relationship.

Response to Review 1

Overall assessment:

This study employs a rather unique approach (a breakdown of the ocean surface energy budget) to gather physical insight on the global precipitation response to warming and to provide an independent observations-based constraint on this response over the recent past. The authors take the approach a step further to place a constraint on the future projected change of global precipitation in models, concluding that models may be overestimating the precipitation sensitivity. Overall, the study is fairly solid (i.e., adequate use of data and simulations), and, to my knowledge, the approach (i.e., employing the ocean surface energy budget) is novel. The study provides some interesting insights, such as the importance of ocean surface albedo for hydrologic sensitivity, and offers evidence that climate model simulations of historical precipitation change are realistic. There are, however, several issues with aspects of the study. The main issues relate to the practical/physical value of the constraint on future precipitation change, physical interpretation of the constraint on historical precipitation sensitivity in models, and the presentation of methods – more details are provided below. If these issues can be adequately addressed by the authors, I feel the paper may be suitable for publication in Nature Communications and has the potential to be a meaningful contribution to the field.

General comments:

1) Constraint on future DP/DT

I am concerned about the practical value and physical significance of the constraint on future precipitation change (DP/DT) that is presented in the Summary section and Table 1. One important aspect that is overlooked is whether the simulated changes during the historical period are statistically and physically linked to projected changes for the future scenarios in models generally. In other words, is there a robust inter-model relationship between the historical and future sensitivities (of K and G terms, and/or their sum) that would support using the observed historical values to constrain the future projected DP/DT? A statistically significant positive correlation between the historical and future scenario(s) sensitivities across models would offer such support. Have the authors looked at such correlations? Based on the results presented in the paper, it almost seems as if the more realistic models (i.e., the sub-ensemble) tend to have higher historical DP/DT values but lower future DP/DT (comparing Fig. 4a and Table 1, for example) – potentially suggesting a negative cross-model correlation between the historical and future scenario K/G term sensitivities to warming. If such negative relationships indeed exist, how can they be explained physically? If no relationships exist, it could suggest that historical changes are fundamentally distinct from future changes (e.g., different forcing agents and/or physical mechanisms) or that the sample size is inadequate, both of which would substantially limit the value of constraining the future P change with the method presented here.

Related to the above is the somewhat unsatisfying spread among the sub-ensemble, with a STD that is often as large or larger than that of the full ensemble (Table 1). In fact, the range of the sub-ensemble DP/DT values (mean +/- 1 STD) encompasses the mean of the full ensemble for all scenarios, suggesting the constraint on future DP/DT is not statistically meaningful.

I recommend the authors further explore the issues raised above, particularly the inter-model relationships between historical and future scenario sensitivities. If such relationships cannot support the constraint procedure currently presented in the paper, then I suggest the authors go in a different direction regarding this aspect of the paper. This could mean searching for another way to place a constraint on the future DP/DT (such as focusing on just the surface shortwave or G terms separately, either of which could possibly exhibit satisfying inter-model relationships between historical and future scenarios). Alternatively, removing the quantitative constraint for the future scenarios entirely and replacing it with a discussion (which highlights the challenges of such an approach) may be another option.

Thank you very much for these constructive comments. Following your suggestion, we have investigated the K_{\downarrow} & G contributions in historical, future transient and $4\times\text{CO}_2$ scenarios. Except for the G contribution in RCP4.5 and RCP8.5, there is no statistically significant correlation between the historical K_{\downarrow} & G contributions and K_{\downarrow} & G contributions in other scenarios across models. In this regard, the emergent relationship does not offer a statistically meaningful constraint on future P. Instead of trying to constrain future $\Delta P/\Delta T$, we now focus on the mechanistic insights revealed by the relationship. We have removed Table 1. This portion of the text has been modified to:

“Opinions are divided as to whether climate models overestimate⁴⁷ or underestimate future $\Delta P/\Delta T$ (ref⁴⁸). Here, we ranked the CMIP5 models according to how close their historical values of $\Delta K_{\downarrow}/\Delta T$ and $\Delta G/\Delta T$ are to the observed values and analyzed the results of 1/3 of the models that rank closest to these observations. This sub-ensemble mean $\Delta P/\Delta T$ is lower than the whole ensemble mean by 4 % (for $4\times\text{CO}_2$) to 19 % (for RCP2.6). However, this result should be interpreted with caution because the correlations between historical and future contributions from ΔK_{\downarrow} and ΔG across models are statistically insignificant (R in the range -0.07 for RCP4.6 to 0.18 for $4\times\text{CO}_2$; $p > 0.05$). The lack of good correlation suggests that mechanisms that change the surface energy balance may be different between historical and future climates. For example, according to the CMIP5 models, a unit rise in temperature results in less surface solar dimming in the future than in the past (Figure 3e) despite a similar rate of water vapor buildup of about 7 % K^{-1} (ref^{3, 27-29}), in part due to differences in aerosols³⁰. It appears that models that are more realistic for the historical climate do not necessarily perform better for future climates.” (L254)

Please also refer to point 2 below.

2) Interpretation/justification for K-G constraint

The physical interpretation of the sum of the surface shortwave (K) and G terms of Eq. 4, used to constrain the historical P sensitivity in models (Fig. 4), is not adequately addressed in the paper. For example, it is not entirely clear why this combination of terms evidently yields the best correlation with DP/DT across models. Several related questions arise: How strong is the relationship between DP/DT and the sum of surface longwave and shortwave components (the latter which are anti-correlated as described in the manuscript)? Is G itself correlated with the surface longwave component? Is an observational constraint that is based on G and surface SW fluxes (as presented here) necessarily more accurate/reliable than one that includes longwave fluxes?

Precipitation temperature sensitivity is positively correlated with the K_{\downarrow} component (Figure 3f), and negatively correlated with the L_{\downarrow} component (Figure 3h). It is positively correlated with the sum of the two components, but the correlation coefficient ($R = 0.776$) is actually lower than the $\Delta P/\Delta T$ versus $\Delta K_{\downarrow}/\Delta T$ correlation ($R = 0.833$). The correlation between $\Delta G/\Delta T$ and $\Delta L_{\downarrow}/\Delta T$ is poor ($R = 0.058$).

We have added a figure with L_{\downarrow} as a controlling variable (Figure 4c) and the following text:

“The relationship in Figure 4a reveals additional diagnostic insights regarding the energy constraints on global P . It suggests that strong compensatory behaviors exist among thermodynamic processes in the climate system. For example, warming and moistening of the atmosphere give rise to predictable increases in L_{\downarrow} (ref^{9, 10}), but because L_{\downarrow} and K_{\downarrow} are tightly coupled (Figure S3), inclusion of the L_{\downarrow} contribution does not bring much improvement to the relationship except for rectifying one outlier (Figure 4c). (The increase in R^2 is marginal, from 0.910 in Figure 4a to 0.912 in Figure 4c.) ...” (L227)

Regarding the physical interpretation, one could argue that G effectively represents the top-of-atmosphere radiative imbalance (i.e., if one reconciles the surface and atmospheric energy budgets mathematically). Indeed, the authors hint at this when describing characteristics of the 4xCO2 scenario on page 8. With this interpretation, the sum of the K and G terms (i.e., K term minus G term) would equate to, I think, a sum including the atmospheric shortwave absorption and outgoing top-of-atmosphere longwave radiation – essentially the atmospheric energy budget excluding the surface sensible heat and surface longwave components. Is this why the correlation in Fig. 4 is so high, because it is dominated by shortwave absorption (a known source of model spread)? Can one then obtain another independent observational constraint using measurements of the shortwave radiative energy budget and outgoing longwave radiation instead?

We agree with the reviewer on this interpretation. In principle, we could use the shortwave radiation budget of the atmosphere and the TOA outgoing longwave radiation as an observational constraint on P . In practice, satellite measurement of the TOA outgoing longwave

is still highly uncertain. It is more feasible to develop this constraint with measurements of the surface energy components, that is, the surface K_{\downarrow} and the ocean heat storage. We have added the following physical explanation of this relationship:

“Since ΔK_{\downarrow} and ΔG are approximately equal to changes in atmospheric absorption of shortwave and outgoing longwave at the TOA, respectively, a physical interpretation of the emergent relationship in Figure 4a is that shortwave absorption (a known source of model spread⁶) and longwave loss at the TOA dominate the global P change” (L221)

“That $\Delta G/\Delta T$ emerges as a dominant control of $\Delta P/\Delta T$ supports the view that monitoring the ocean heat content could be the best strategy available to constrain future P change³. Since global dimming is the other dominant contributor, long-term monitoring of solar radiation at the earth’s surface, especially at marine locations, should provide another strong constraint on P .” (L234)

Prompted by your comment, we have explored the physical meaning of the relationship between $\Delta P/\Delta T$ and the G component. The results are included in two new Figures (Figures 5 and S7) and the interpretation in a new paragraph:

“Figure 4a implies a connection between the P temperature sensitivity and the strength of climate feedback. In the abrupt $4\times\text{CO}_2$ scenario, the TOA radiation imbalance decreases and the surface temperature increases over time after the sudden CO_2 rise. In the paradigm of radiative forcing versus climate feedback, the slope of the TOA radiation imbalance versus surface air temperature is a measure of the feedback strength⁴⁶. Since G accounts for a great majority of the imbalance, the magnitude of $\Delta G/\Delta T$ obtained from $4\times\text{CO}_2$ simulations can be regarded as a good approximation of the feedback strength. We find that among the CMIP5 ensemble of models, those with a stronger feedback strength tend to give a higher $\Delta P/\Delta T$ in the $4\times\text{CO}_2$ scenario ($R = 0.41$, $p < 0.05$; Figure 5a). This positive correlation between the hydrological climate sensitivity and the feedback strength is also evident from simulations with one CMIP5 model member (MIROC5) under different states of perturbed ocean evaporation⁴⁷. The feedback strength on its own, however, has a limited ability of explaining inter-model variations for the historical climate and for future transient scenarios (Figure S7)” (L240)

In summary, the energy-based constraint on historical $\Delta P/\Delta T$ in models (Fig. 4) is presented in a way that is more mathematical than physical and that raises a number of unresolved questions (some examples of those questions given above). I think this component of the paper would benefit from a more thorough discussion that includes more physical explanation, justification of the approach, and discussion of potential advantages of this approach versus other similar or equivalent approaches. This may potentially require additional calculations.

Thank you for these constructive comments. We have added a discussion section on these points. In this revision, physical explanation is given wherever appropriate. The justification for our

approach and its advantages over other approaches are explained in the introduction (L87-94). We have also clarified the relationship between our diagnostic method and a similar method used by Siler et al. (point 3, review 3).

3) Methods not adequately described

A key theme of this paper is using observations to place constraints on DP/DT. Yet, many details of the methods for obtaining observational estimates and their uncertainties are glossed over. Additionally, some calculations from the models are not adequately described. The lack of information would make reproducing (or expanding upon) the results difficult if not impossible for the research community, and also makes it hard to judge potential shortcomings of the approach. Specific examples are given below, and should be addressed by adding more details to the text and/or tables in the appropriate places:

- Table S1: It is not clear how the model s values are obtained. Is it done the same way as for MERRA-2 (i.e., a regression across years) or a different way? A follow-up question is: exactly how are the standard deviations computed, in either case?

We have added these details to the Table caption:

“For each CMIP scenario, s is the slope of linear regression between changes in global precipitation (ΔP) and ocean evaporation (ΔE_o) across models (with intercept forced through zero), where ΔP and ΔE_o are differences in global precipitation and ocean evaporation, respectively, between the last and the first 10-years of each model simulation. For MERRA-2, s is the slope of linear regression between annual global P and global E_o (with intercept forced through zero). Uncertainty range is \pm one standard deviation, estimated as half of the 95% confidence bound on the regression slope. n – number of models (climate scenarios) or number of years (reanalysis); ϕ – land modifier, the ratio of land evaporation change to ocean evaporation change; R – linear regression coefficient. All correlations are significant at $p < 0.001$.” (Table S1)

- How is G computed in models?

We have added the following description:

“The G term is the net heat flux entering the liquid water column plus a small amount of energy consumption due to ice melt at high latitudes. In the above diagnostic analysis, G was calculated as the residual of the ocean surface energy balance equation $G = R_n - H - \lambda E_o$ (Figure S4). The surface net radiation (R_n) and the ocean sensible (H) and latent heat flux (λE_o) were obtained from the atmospheric dataset archived for each model simulation. This residual calculation ensures that energy is conserved in our diagnostic analysis.” (L476)

More information about the model scenarios is warranted. For example, what does ssp585 mean and how does it compare qualitatively with the CMIP5 scenarios? Which years specifically are

used from each scenario? Perhaps add this information to an existing table (S1?) or create a new table.

We now give a brief description of ssp585 in the Methods section:

“Scenario ssp585 is an energy and resource intensive socioeconomic scenario for the 21st century resulting in a similar 2100 radiative forcing (8.5 W m^{-2}) as its CMIP5 predecessor RCP8.5.” (L464)

Information on simulation periods is added to the Table caption:

“The CMIP5 simulation periods for historical, future (RCP2.6, RCP 4.5, RCP6.0 and RCP8.5) and $4\times\text{CO}_2$ scenarios are 1850 – 2005, 2006 – 2100 and 1850 – 1999, respectively. For CMIP6 ssp5-8.5, the simulation period is 2015-2100.” (Table S3)

- Regarding methods for observational estimates (starting L292): It appears the estimates are in part taken from references and in part derived by the authors, but the degree of each is not always clear. For example, is Beta purely taken from ref. 12? How much of the G calculation was already done by ref. 37 and how much was done by the authors? More details about the procedures for deriving Beta and G are also warranted for reproducibility, especially considering the importance of G for constraining DP/DT in models.

This part of the Methods has been expanded to

“(b) *Ocean Bowen ratio* (β): according to the modified version of the Priestley-Taylor model on the basis of the Objectively Analyzed Air-sea Flux dataset¹², oceanic β is inversely proportional to the slope of the saturation vapor pressure versus temperature T . The β temperature sensitivity was obtained from the derivative of this function with respect to T and evaluated at the observed global mean temperature. Its uncertainty was based on the spread of observed historical temperature.” (L404)

“(f) *Ocean heat storage* G : $\Delta G/\Delta T$ was obtained by a quadratic fit of the ocean heat content³⁷ (OHC) against time (t) as $\text{OHC} = a_0 + a_1 t + a_2 t^2$. Dividing the coefficient of the quadratic term a_2 by the ocean area gave the time rate of change of the heat flux into the water column, and multiplying this rate by the length of the observational period (1955 to 2017), we obtained ΔG . The uncertainty on ΔG was estimated as $\frac{1}{2}$ of the 95% confidence bound on a_2 . We then estimated $\Delta G/\Delta T$ by dividing ΔG with the temperature change ΔT of 0.774 K observed over the same period according to GISTEMP.” (L425)

- Also regarding observational estimates: Which specific years are used from each of the reanalysis datasets? Is the GISS temperature used for all terms?

The specific years used are now summarized in a new Table, along with detailed statistics (Table S4). We used the global mean temperature from each reanalysis to obtain the K_{\downarrow} and L_{\downarrow} sensitivities.

“... We first established a linear relationship between the annual area-weighted K_{\downarrow} over the ocean grids and the annual mean global 2-m air temperature from the same reanalysis.” (L410)

- Also regarding observational estimates: The quantification of uncertainty is hastily and inadequately described. For example, what constitutes the 1,000,000 ensemble members? More explanation and details are warranted.

We have added the following details:

“The uncertainty of $\Delta P/\Delta T$ was determined with a Monte Carlo method involving 1,000,000 ensemble members. For each member, each term on the right-hand side of Equation (1) was the sum of its mean value (Tables S1 and S2) and a random error produced by a random number generator. This error was assumed to vary independently from other terms and according to a normal distribution with the standard deviation given in Table S1 or S2. The uncertainty of $\Delta P/\Delta T$ was calculated as one standard deviation of the ensemble after the top and bottom 0.5% of outliers were excluded.” (L441)

Other specific comments:

L29: “empirical data about the atmospheric longwave loss are still highly uncertain” - Two recent studies that are relevant to this statement, which are not currently cited in this manuscript, are Su et al. 2017 (<https://www.nature.com/articles/ncomms15771>) and Watanabe et al. 2018 (<https://www.nature.com/articles/s41558-018-0272-0>): independent observational constraints on DP/DT that are based on longwave measurements. The authors should consider incorporating these references into the discussion here.

These two papers (references 47 and 48) are now cited. Thank you.

“This positive correlation between the hydrological climate sensitivity and the feedback strength is also evident from simulations with one CMIP5 model member (MIROC5) under different states of perturbed ocean evaporation⁴⁷.” (L248)

“Opinions are divided as to whether climate models overestimate⁴⁷ or underestimate future $\Delta P/\Delta T$ (ref⁴⁸).” (L254)

“Several mechanisms are known to reduce cloud cover in these regions in a future warmer climate, including breakup of stratocumulus cloud decks⁵¹, aggregation of deep convective clouds⁵¹, and high cloud shrinkage associated with tightening of the ascending branch of the Hadley Circulation⁴⁸.” (L284)

L66: A reference to Methods/equation 1 would be helpful.

Added.

L78: Add reference to Methods/equation 4.

This equation is now moved to the main text here, following the suggestion by Reviewer 2.

L94: It's rather important to add a reference to the Methods here.

Added.

L99-100: “The ϕ value is lower according to the reanalysis data (-0.05) than the ensemble model mean for the historical climate (0.15; Table S1)” – If s for the models is estimated as a regression across models (rather than regression across years, as for MERRA-2), could this be why the phi values are different? If so, it would be informative to compute phi in models using a regression across years as well, to not only address this question but also potentially provide another line of evidence for a strong relationship (R) between global P and oceanic E.

We have tried the regression of annual mean P versus annual mean Eo across years using CMIP5 historical simulations. The annual P and Eo are highly correlated across years, yielding a mean R of 0.923 ± 0.055 . The mean regression slope (0.766 ± 0.091) is slightly higher with the slope obtained across models (0.754 ± 0.054 ; Table S1). So the regression procedure does not appear to be the source of the difference between the reanalysis ϕ and the modeled ϕ .

L137-146: What role does water vapor play in this relationship, if any? I think this is worth including in the discussion here, as water vapor appears a few other times throughout the paper in other contexts.

According to the radiation transfer calculated by Pendergrass and Hartmann (2014) for clear sky conditions, the K_{\downarrow} and L_{\downarrow} temperature sensitivity due to atmospheric moistening is -0.9 and $+3.4 \text{ W m}^{-2} \text{ K}^{-1}$, respectively. In other words, atmospheric moistening cannot explain the 2:1 relationship. We have added their result in Figure S3 (open white square) and have changed the text to:

“Three mechanisms are known to cause a negative relationship between ΔK_{\downarrow} and ΔL_{\downarrow} . Atmospheric moistening at higher temperatures reduces K_{\downarrow} slightly and increases L_{\downarrow} by four times as much¹⁰ (Figure S3)...” (L148)

L294-299: This information regarding the computation of Dalpha/DT should be included, to a reduced extent, in the Fig. S1 caption, since the reader will likely look at Fig. S1 before reading this part of the Methods.

Done. Thank you for this suggestion.

L312-313: “Assuming $\Delta T = 3$ K at CO₂ doubling, this gives an additional sensitivity of 0.46 W m⁻² K⁻¹.” As demonstrated in ref. 31, this value of warming at CO₂ doubling is not entirely known and is unconstrained. I therefore recommend incorporating additional uncertainty into this component.

Excellent point. We have incorporated this uncertainty into the longwave component.

Acknowledgements/ data statement: I do not see an acknowledgements section stating where the data used for the paper can be accessed. Such a section is important for reproducibility.

Data availability statement is now added.

Response to Review 2

I would like to acknowledge the authors' work, the quality of the text and the effort to provide a different perspective on constraining global rainfall with a different argument. I recommend for publication with minor revisions. Some comments are mostly semantic in order to improve clarity, and some include a few questions on the results and the sources of uncertainty. A more thorough discussion on physical link between this new constraint and the standard one based on atmospheric energy balance could also strengthen the manuscript.

Thank you for these encouraging comments. We have added a discussion section on the physical link between the surface energy constraint and atmospheric energy processes.

• L13: please clarify the term “driver”; by “drivers of dP/dT ” do you mean “source of intermodel spread in dP/dT ”? If instead you mean “driver of change in P ”, then ocean albedo mentioned just above also seems to be an important driver. In general throughout the paper, it would be helpful to make a distinction between what is responsible for the change dP/dT (the drivers) and what is responsible for the spread across models or additional sources of uncertainty missing from models. I agree that in the spirit of Allen and Ingram (2002), both are interchangeable, but it seems like it is not the case in your study.

We used the term “driver”, in a narrow sense, to describe how changes -- either physical or due to model error -- in a surface energy component contribute to the $\Delta P/\Delta T$ variability. In the broad climate change literature, this term is often used to describe agents of climate change. To avoid confusion, in the revision we have replaced it with “control”. For example, this sentence (in the abstract) is now changed to:

“In this surface energy balance framework, the incident shortwave radiation at the ocean surface and the ocean heat storage exert a dominant control on \$\Delta P/\Delta T\$, explaining 91% of the Assessment Report 5 inter-model spread and the spread across climate scenarios.” (L13)

• L15: clarify why the high bias of “4% to 19%” does not match the original discrepancy in the dP/dT found to be “3-4 times smaller than the rates projected” (L9). Just by reading the abstract, it is not clear whether these two statements are supposed to match; it seems so, as it seems that L9 was a motivating question. Is dP/dT still 3-4 times smaller in the historical runs than in the future transient runs also when looking at the sub-ensemble (L198-199)? If so, what explains the remaining difference between the historical dP/dT and future dP/dT ?

The “4% to 19%” bias is the difference in future $\Delta P/\Delta T$ between the whole ensemble and a sub-ensemble of the CMIP5 models. In response to Reviewer 1’s suggestion (point 1, review 1), we

have now removed this from the abstract. We have also removed the wording “3-4 times smaller than the rates projected” on the suggestion by Reviewer 3.

• **L26: “the atmosphere will lose more LW radiation [...] as it accumulates CO₂” is correct, but it seems necessary to mention that it is also largely due to water vapor effects in the LW: the next sentence highlights the opposing effect water vapor in the SW, and as currently stated the paragraph implicitly suggests that CO₂ and H₂O oppose each other, which is a little bit too simplistic.**

This is a good point. This sentence has been modified to
“The atmosphere will lose more longwave radiation energy to the outer space and to the Earth’s surface as its temperature increases due to rising CO₂ and as it accumulates water vapor.” (L26)

• **L29-30: that is a good argument;**

1. Pendergrass and Hartmann (2014) also argue that P is more strongly constrained by the atmospheric energy budget than the surface energy budget because of the equilibration time scale of the ocean being larger. Not sure this point is totally relevant, but maybe you could discuss this time scale question here, as well as how it affects your results? For instance in the 4×CO₂ run, which shows an energy imbalance in the ocean.

In this revision, we have clarified that our surface energy balance approach is used to diagnose the slow response of P to rising temperature. The fast P adjustment is not considered. In this regard, the atmospheric energy balance approach is more advantageous because it can be used to diagnose both the total change and the slow change in P, as demonstrated by Pendergrass and Hartmann (2014), DeAngelis et al. (2015), Fläschner et al. (2016), Siler et al. (2019), and others.

“In the 4×CO₂ scenario, $\Delta P/\Delta T$ is equivalent to the hydrological sensitivity parameter defined by Fläschner et al.⁵ and represents the slow response of P to warming (Figure S8). The P response to warming analyzed by Siler et al.¹⁶ is similar to the apparent hydrological sensitivity given by Fläschner et al.⁵. Fast P adjustment, taken as the y intercept of the P versus temperature regression for the 4×CO₂ simulation in reference to piControl⁵, and fast P response (the P difference between sstClim and sstClim4×CO₂ simulations¹⁶) are not considered in this study.” (L468)

2. “it is not possible to estimate the P change as a residual of the atmospheric energy balance”; true, but from your results, it is not possible to estimate it from a residual in the ocean energy balance either, unless we have reliable estimates of ocean heat uptake, correct? Maybe the importance of G (and whether or not we have confidence on its estimates) could be emphasized in the conclusion a bit more?

We actually used the observed ocean heat uptake to close the surface energy balance and to get an estimate of the historical $\Delta P/\Delta T$. The observational uncertainty on this term is smaller than the

uncertainty on the incoming longwave and incoming shortwave radiation (Table S2). This portion of the Methods has been rewritten as:

“(f) *Ocean heat storage G*: $\Delta G/\Delta T$ was obtained by a quadratic fit of the ocean heat content³⁷ (OHC) against time (t) as $OHC = a_0 + a_1t + a_2t^2$. Dividing the coefficient of the quadratic term a_2 by the ocean area gave the time rate of change of the heat flux into the water column, and multiplying this rate by the length of the observational period (1955 to 2017), we obtained ΔG . The uncertainty on ΔG was estimated as $\frac{1}{2}$ of the 95% confidence bound on a_2 . We then estimated $\Delta G/\Delta T$ by dividing ΔG with the temperature change ΔT of 0.774 K observed over the same period according to GISTEMP.” (L425)

In response to your suggestion, we have added the following sentence to the abstract:

“On the other hand, the observed increase in ocean heat storage weakens the historical P .” (L12)

The role of G is also emphasized in the (new) Discussion section

“That $\Delta G/\Delta T$ emerges as a dominant control of $\Delta P/\Delta T$ supports the view that monitoring the ocean heat content could be the best strategy available to constrain future P change³.” (L235)

• L45 and L63-73: the contributions come from the ocean energy balance, and the estimation of the uncertainty as well. Could you estimate the role of land use change, land drying/wild fires and changes in vegetation/desertification on the uncertainty in dP/dT for future climates? How would ϕ affect the uncertainty range provided L188? Would we get an error range that is larger than the one obtained from the atmospheric budget argument?

The CMIP5 modeling result (Figure S2) suggests that decrease in forest cover, such as via wildfires and desertification, would reduce ϕ or the role of land evaporation on global P . We noted that ϕ is lower according to MERRA-2 reanalysis than that from the CMIP5 historical simulations (L109; Table S1). Use of the climate model mean ϕ would increase $\Delta P/\Delta T$ by only 8% (L113). The estimate of $\Delta P/\Delta T$ on L188 (now L195) is not affected by ϕ because it comes from the emergent relationship between the modeled $\Delta P/\Delta T$ versus incoming solar radiation and ocean heat storage.

• L84-85: I am not familiar with the Priestley-Taylor model. Could you state the physical reason for the change in Bowen ratio with warming? Why would it be more efficient to have a larger fraction of surface cooling by increasing the contribution from evaporation than that of dry turbulent fluxes?

We have expanded the text here:

“An advantage of performing energy balance analysis over the oceans rather than over the whole globe is that ocean evaporation occurs at the potential rate limited by energy only, whereas land evaporation is confounded by both soil moisture and energy availability and is more difficult to determine from observational data. For this reason, the ocean β can be determined with the classic Priestley-Taylor model of potential evaporation. As temperature rises, the vapor pressure

at the water surface increases exponentially according to the Clausius–Clapeyron equation. This results in a faster change in the sea-air vapor pressure gradient than in the temperature gradient, and β decreases¹⁶.” (L87)

• **L95-97: please clarify whether these two paths (effect of sea-ice melting on warming and on precipitation) are actually physically distinct and what makes them distinct. In both cases precipitation increases because of an increased latent heat flux from the surface (either via a T increase or via an albedo decrease which allows for more surface absorption). The question is actually : are you referring to P itself or dP/dT? Is dP/dT physically independent of T? It seems that by “melting of sea ice amplifies warming” you refer to the direct heating effect of downwelling radiation at the surface, and by “melting of sea ice also intensifies precipitation” you are referring to some additional/marginally increasing fraction of the absorbed radiation that is converted into latent heat, which is not P but dP/dT. Am I correct?**

You are correct. We are referring to albedo contribution to $\Delta P/\Delta T$. This sentence has been changed to:

“Our result shows that the same process also increases the global precipitation temperature sensitivity.” (L106)

• **L146: this would be interesting to discuss the role of low clouds on the uncertainty itself, potentially in the conclusion where you also mention low clouds. You mention they are the key to explain the relationship between dK and dL, so they are key to argue that dK is an important driver of dP. But in parallel, we know the dynamics of low clouds can change with warming.**

The dynamics of low clouds are increasingly recognized to play a crucial role in modeling the equilibrium climate sensitivity. Watanabe et al. (ref 47) have investigated their role in influencing the P temperature sensitivity. We hope that our paper can serve as another impetus to this line of investigation. We have added the following text to the Discussion section:

“At low latitudes, cloud cover change can also influence P . Climate models with a higher equilibrium climate sensitivity are shown to have a more positive low-cloud feedback³¹ and agree better with constraints provided by the cloud behaviors observed in tropical and subtropical oceans than lower sensitivity models³²⁻³⁴. Several mechanisms are known to reduce cloud cover in these regions in a future warmer climate, including breakup of stratocumulus cloud decks⁵¹, aggregation of deep convective clouds⁵¹, and high cloud shrinkage associated with tightening of the ascending branch of the Hadley Circulation⁴⁸. The ocean surface K_1 will increase in response to the reduction in cloud cover, but it is not known if this increase is large enough to offset the dimming caused by rapid water vapor buildup in the tropical and subtropical atmosphere so as to result in a net increase in P . Numerical perturbation experiments may be necessary to disentangle the role of these interactive regional processes in the global P response.” (L280)

- **L287: it would be helpful to see equation (4) in the main text (or some simplified form of it).**

Done

- **L345-346: do the atmospheric constraint and the ocean constraint give the same estimate for the run 4×CO₂ (see comment made for L29-30)?**

Yes, the two constraints are in very good agreement. (The atmospheric result is based on Figure 6a of Fläschner et al., ref 5.) We have added the following sentence to the Methods section:

“The offline $\Delta P/\Delta T$ from the surface energy balance (Equation 1) for the 4×CO₂ scenario ($2.18 \pm 0.21 \text{ W m}^{-2} \text{ K}^{-1}$; Figure 2g) agrees well with the $\Delta P/\Delta T$ diagnosed from the atmospheric energy balance ($2.03 \text{ W m}^{-2} \text{ K}^{-1}$; ref⁵).” (L490)

--- **References**

Myles R. Allen and William J. Ingram. Constraints on future changes in climate and the hydrologic cycle. *Nature*, 419(6903), 2002. ISSN 00280836. doi: 10.1038/nature01092.

Angeline G Pendergrass and Dennis L Hartmann. The Atmospheric Energy Constraint on Global-Mean Precipitation Change. *Journal of Climate*, 27(2):757–768, jan 2014. ISSN 0894-8755. doi: 10.1175/JCLI-D-13-00163.1. URL <http://journals.ametsoc.org/doi/abs/10.1175/JCLI-D-13-00163.1>.

Response to Review 3

In this paper, the authors use the surface energy budget to decompose the change in global precipitation with global warming into contributions from changes in albedo, the Bowen ratio, net surface radiation, and ocean heat storage. Within this framework, they find a significant role for changes in ice albedo, contradicting earlier studies. They find that shortwave radiation and ocean heat storage account for a large fraction of the intermodal spread. They then propose an emergent constraint on global hydrologic sensitivity to climate change based on observed changes in shortwave radiation and ocean heat storage.

While there are aspects of the study that I find interesting and valuable, I think it has a few serious problems that should prevent it from being published.

1. I'm skeptical that the decomposition in Eq. 4 can be applied in the global mean in a way that's physically meaningful. The reason is that the Bowen ratio is generally smaller where temperatures are warm. Thus, the response of evaporation to changes in radiation or ocean heat storage is quite sensitive to where/when those changes occur: If they occur in regions or seasons in which the Bowen ratio is large (e.g., at high latitudes), their contribution to evaporation change will be quite small.

I suspect this explains why the authors find a large role for changes in ice albedo, in contrast to previous studies. In Eq. 4, the efficacy of albedo change is controlled by $1/(1+\beta)$. Since albedo changes are concentrated at high latitudes, it would be appropriate to use the value of β at these latitudes. By instead using the (smaller) global-mean value of β , the contribution from the change in albedo is likely exaggerated. This would also explain the compensation between changes in albedo and changes in the Bowen ratio, as noted in lines 116-117.

Thank you for this insightful criticism. Four concerns are imbedded in this comment, regarding the role of albedo, global-scale analysis, physical interpretation, and compensation effect/methodological limitation. Please allow us to respond to each.

The role of albedo

It is true that β is higher at lower temperatures. The β for high-latitude ocean (north of 60° N and south of 60° S) is about 0.70 according the modified Priestley-Taylor model of Yang and Roderick (ref 12), or about four times the global ocean mean β (0.16; Table S2). In our analysis, albedo a is calculated as the ratio of the annual mean outgoing shortwave radiation over all the ocean grids to the incoming shortwave radiation over these grids. The a temperature sensitivity (-0.0065 K^{-1} ; Table S2) is therefore the global mean value. If we restrict the analysis to high-latitude grids, the a temperature sensitivity would increase by >10 times to 0.087 K^{-1} (Figure R1 below), which would be more than enough to compensate for the high β value. Because all the other terms of the surface energy balance (including β) are expressed as global-scale values, we

feel that it is more appropriate to use the global value than the regional value to isolate the albedo contribution to the P change.

Figure R1: Albedo temperature sensitivity according to the CERES observation: open circles, polar ocean (north of 60° N and south of 60° S); solid circles, global ocean. (The same global data is also given in Figures S1 using a smaller scale range.)

In this revision we have clarified how α and β are calculated in our diagnostic analysis: “ In the diagnostic analysis presented above, the energy fluxes in Equation (4) are area-weighted ocean mean values, α is the ratio of area-weighted mean reflected to incoming solar radiation, and β is the ratio of area-weighted mean sensible to latent heat flux. In addition, the lateral transport of heat via ocean currents is zero at the global scale. For these reasons, Equation (4) is exact at the global scale.” (L372)

Why global scale

The global energy balance framework used in this study is not new. Like many other published studies, we chose this framework because of its well-known strength: at the global scale, energy balance provides a strong thermodynamic constraint on the hydrological cycle. This constraint has been examined from both the atmospheric (e.g., Allen and Ingram, ref 3) and the surface energy perspective (e.g., Siler et al, ref 16). At regional scales, the power of atmospheric energy balance for constraining the hydrological cycle is much weaker because of lateral advection of energy into the regional domain of interest. Likewise, transport of energy by ocean currents would complicate the surface energy balance analysis at regional scales.

A methodological novelty of this study is that we used this global framework to provide an estimate of the historical $\Delta P/\Delta T$ from surface observations. We believe that this is a useful contribution to the published literature.

Physically meaningful interpretation

The reviewer questions whether the global energy balance analysis can produce physically meaningful results. We acknowledge that this aspect of our original submission is not strong. In this revision, we have added physical interpretation wherever appropriate. Here, we give a summary of these additions:

- Comparison with atmospheric energy balance: “Our results based on the surface energy consideration can be put into the context of atmospheric energy conservation. ... Since ΔK_{\downarrow} and ΔG are approximately equal to changes in atmospheric absorption of shortwave and outgoing longwave at the TOA, respectively, a physical interpretation of the emergent relationship in Figure 4a is that shortwave absorption (a known source of model spread⁶) and longwave loss at the TOA dominate the modeled P change.” (L199)
- Utility of the emergent relationship: “The relationship in Figure 4a reveals additional diagnostic insights regarding the energy constraints on global P ... That $\Delta G/\Delta T$ emerges as a dominant control of $\Delta P/\Delta T$ supports the view that monitoring the ocean heat content could be the best strategy available to constrain future P change³. Since global dimming is the other dominant contributor, long-term monitoring of solar radiation at the earth’s surface, especially at marine locations, should provide another strong constraint on P .” (L227)
- Connection to climate feedback strength: “Figure 4a implies a connection between the P temperature sensitivity and the strength of climate feedback. In the abrupt $4\times\text{CO}_2$ scenario, the TOA radiation imbalance decreases and the surface temperature increases over time after the sudden CO_2 rise. In the paradigm of radiative forcing versus climate feedback, the slope of the TOA radiation imbalance versus surface air temperature is a measure of the feedback strength⁴⁶. Since G accounts for a great majority of the imbalance, the magnitude of $\Delta G/\Delta T$ obtained from $4\times\text{CO}_2$ simulations can be regarded as a good approximation of the feedback strength. We find that among the CMIP5 ensemble of models, those with a stronger feedback strength tend to give a higher $\Delta P/\Delta T$ in the $4\times\text{CO}_2$ scenario ($R = 0.41$, $p < 0.05$; Figure 5a). This positive correlation between the hydrological climate sensitivity and the feedback strength is also evident from simulations with one CMIP5 model member (MIROC5) under different states of perturbed ocean evaporation⁴⁷. The feedback strength on its own, however, has a limited ability of explaining inter-model variations for the historical climate and for future transient scenarios (Figure S7).” (L240)

Compensation effect/methodological limitation

The emergent relationship (Figure 4a) and diagnostic results (Figure 2) are outcomes of strong compensating behaviors – including interactions between α and β as pointed out by the reviewer – among thermodynamic processes of the climate system. However, the global diagnostic framework itself does not provide a clear mechanistic understanding of the nature of these behaviors. This limitation is now acknowledged in the revision.

“Our diagnostic analysis (via Equation 1) is restricted to the global scale. Even though it has shed new light on the manifestation of interactions among energy variables, a mechanistic understanding of the nature of these interactions will require more granular examination at local

and regional levels. Rising temperatures will decrease ocean β (ref^{12, 16}). Since β is already very low for mid- to low-latitude ocean regions (about 0.13 between 60° S and 60° N), this thermodynamic response is more important for high-latitude regions where the high β (about 0.70 north of 60° N and south of 60° S) allows more room for energy allocation shift from sensible heat to latent heat as evident in historical climate simulations⁴³. On the other hand, the high β may counteract the increase of radiation energy available for evaporation via a reduction in polar waters. Additionally, changes in K_{\downarrow} and a at high latitudes are positively correlated in the CERES data and across the CMIP5 models (Figure S6), consistent with the observation of greater cloud cover during low-ice years near the North Pole⁴⁹. Thus, change in regional K_{\downarrow} is another process that may counteract the albedo effect on global P . At low latitudes, cloud cover change can also influence P . Climate models with a higher equilibrium climate sensitivity are shown to have a more positive low-cloud feedback³¹ and agree better with constraints provided by the cloud behaviors observed in tropical and subtropical oceans than lower sensitivity models³²⁻³⁴. Several mechanisms are known to reduce cloud cover in these regions in a future warmer climate, including breakup of stratocumulus cloud decks⁵¹, aggregation of deep convective clouds⁵¹, and high cloud shrinkage associated with tightening of the ascending branch of the Hadley Circulation⁴⁸. The ocean surface K_{\downarrow} will increase in response to the reduction in cloud cover, but it is not known if this increase is large enough to offset the dimming caused by rapid water vapor buildup in the tropical and subtropical atmosphere so as to result in a net increase in P . Numerical perturbation experiments may be necessary to disentangle the role of these interactive regional processes in the global P response.” (L268)

2. The authors lump together the direct effects of CO2 and warming, but I think it’s important to think of these as separate. By itself, an increase in CO2 causes a decrease in evaporation through an increase in ocean heat storage. This is sometimes called the “fast” response to CO2, since it is not mediated by temperature change. The “slow”, or temperature-mediated response represents the direct impact of warming. The way the authors define the changes in each variable (last 10 years minus first 10 years) doesn’t distinguish between these effects, so it’s hard to understand what’s going on.

This is an excellent point. In this revision, we have clarified that the precipitation change in our diagnostic analysis is the temperature-mediated slow response. We have added a figure (Figure S8) to demonstrate this.

“In the 4×CO₂ scenario, $\Delta P/\Delta T$ is equivalent to the hydrological sensitivity parameter defined by Fläschner et al.⁵ and represents the slow response of P to warming (Figure S8). The P response to warming analyzed by Siler et al.¹⁶ is similar to the apparent hydrological sensitivity given by Fläschner et al.⁵. Fast P adjustment, taken as the y intercept of the P versus temperature regression for the 4×CO₂ simulation in reference to piControl⁵, and fast P response (the P difference between sstClim and sstClim4×CO₂ simulations¹⁶) are not considered in this study.” (L468)

As pointed out by the reviewer, Allen & Ingram (ref 3) and others, in the absence of any tropospheric (temperature) changes, global P will decrease in response to increase in CO₂. This fast P response is well-understood from the atmospheric energy balance perspective. However, we lack consensus on how to best explain this behavior from the surface energy balance perspective. The reviewer suggests that it is caused by the change in ocean heat storage G. Here, we prefer to interpret the G change as a slow response because the change occurs at a long timescale and is a temperature-mediated process linked to climate feedback (point 2, review 1; point L29-30, review 2). Kamae et al. (2015, Curr Clim Change Rep 1:103–113) argue that moistening of the near-surface air over the ocean is the mechanism responsible for the slowdown of ocean evaporation and the hydrological cycle immediately after a sudden rise in atmospheric CO₂. Using MIROC5 AGCM simulations, they showed that the timescale of this adjustment is on the order of several days.

A good example of why this is a problem is evident in Fig. 3h, which shows the change in P vs. the change in longwave radiation. There's a clear difference between the inter-ensemble regression slope, which is negative, and the regression slope within a given ensemble, which is harder to discern, but appears to be slightly positive. I'm guessing the overall negative slope is mostly driven by differences in forcing, which don't exist within a given ensemble.

The reviewer is correct that the relationship of the P change vs the L_↓ change is different between inter-model variations and inter-scenario variations. Similar behaviors also exist for other energy balance variables. For example, the P change and the Bowen ratio contribution are negatively correlated across scenarios (Figure 3c) but show a positive correlation within some scenarios (e. g, R = 0.28 for 4×CO₂). This inconsistency reveals a weakness of the single-variable analysis as was done in some previous studies. A key result of our study is that the combined contribution from the K_↓ and G changes can explain both the inter-model and the inter-scenario variations. We have added the following text to further explain this result:

“...However, when examined individually, these energy components generally lack consistency between within-scenario and inter-scenario variations. For example, the relationship between L_↓ change and $\Delta P/\Delta T$ is positive for the 4×CO₂ scenario ($R = 0.25$) but is negative across scenarios (Figure 3h). In contrast, consistency is achieved if the incoming shortwave at the ocean surface and the ocean heat storage are combined (Figure 4a). Since ΔK_{\downarrow} and ΔG are approximately equal to changes in atmospheric absorption of shortwave and outgoing longwave at the TOA, respectively, a physical interpretation of the emergent relationship in Figure 4a is that shortwave absorption (a known source of model spread⁶) and longwave loss at the TOA dominate the modeled P change.” (L217)

“The relationship in Figure 4a ... suggests that strong compensatory behaviors exist among thermodynamic processes in the climate system. For example, warming and moistening of the atmosphere give rise to predictable increases in L_↓ (ref^{9, 10}), but because L_↓ and K_↓ are tightly coupled (Figure S3), inclusion of the L_↓ contribution does not bring much improvement to the

relationship except for rectifying one outlier (Figure 4c). (The increase in R^2 is marginal, from 0.910 in Figure 4a to 0.912 in Figure 4c.)” (L227)

According to Stephens & Hu (ref 9) and Pendergrass & Hartmann (ref 10), the $L\downarrow$ change in this study is a temperature-mediated response. They showed that under clear-sky conditions, about half of $\Delta L\downarrow$ is attributed to warming of the atmosphere and the other half to water vapor buildup in a warmer atmosphere.

Similarly, the change in ocean heat storage is quite sensitive both to CO2 forcing and to atmospheric warming, since it is roughly equal to the net radiation imbalance at the top of the atmosphere. We don’t know what’s going on physically when these effects are lumped together.

Reviewer 1 made a similar comment regarding the role of G. Please refer to our response to your comment 1 above (“Physically meaningful interpretation”) and to point 2, review 1.

3. The authors don’t sufficiently engage with previous work. For example, Siler et al. (2019) perform a similar decomposition derived from the Penman equation. The authors should address how their work differs from and builds on this work and other related decompositions based on the atmospheric and surface energy budgets.

We cited this paper (ref 16) to support the argument that the global P change is primarily driven by the change in ocean evaporation (L59). In this revision, we have offered a more detailed explanation of the relationship of this study to the study by Siler et al.

“As temperature rises, the vapor pressure at the water surface increases exponentially according to the Clausius–Clapeyron equation. This results in a faster change in the sea-air vapor pressure gradient than in the temperature gradient, and β decreases¹⁶.” (L93)

“... Rising temperatures will decrease ocean β (ref^{12, 16})...” (L271)

“Our analytical framework can be considered an extension of the work by Siler et al.¹⁶ who decomposed future P change with the ocean surface energy balance equation. In their study, the thermodynamic response, or shift of energy allocation from sensible heat to latent heat, consists of change in the equilibrium Bowen ratio and changes in boundary layer dynamics/relative humidity. It can be shown that their diagnostic equation (their Equation 16, without the boundary layer term) is identical in form to the terms in the curly brackets of Equation (1). In this study, the thermodynamic response is determined with the Bowen ratio from the modified Priestley-Taylor model of ocean evaporation¹² and the actual Bowen ratio from sensible heat and latent heat fluxes calculated by climate models. Because the actual Bowen ratio is less sensitive to temperature than the theoretical equilibrium Bowen ratio, this thermodynamic contribution to the global P change is smaller in our assessment. Additionally, we have introduced a land modifier to account for the land evaporation contribution to global P .” (L382)

In response to this comment and the comments made by Reviewers 1 and 2, we have added a new section (Discussion) where we interpret the key results of our diagnostic analysis in the context of the published literature. (Thank you for your constructive suggestion.)

Line-by-line comments

Abstract: “find that historical warming intensified P at a rate of 0.39 ± 0.40 %/K, which is ~3-4 times smaller than the rates projected for future transient climates”: The authors seem to be implying that future projections are inconsistent with observations, but that’s not necessarily true. The sensitivity should be larger the closer the climate is to equilibrium.

Thank you for sharing your insights with us. In response, we have changed this sentence to: “Here, using observations of the ocean surface energy balance as a new hydrological constraint, we find that historical warming intensified P at a rate of 0.60 ± 0.44 % K^{-1} , which is slightly higher than the multi-model mean calculation for the historical climate (0.38 ± 1.18 % K^{-1})” (L7)

25. “The atmosphere will lose more longwave radiation energy to the outer space ... as it accumulates CO₂.” This is wrong: increasing CO₂ causes a reduction in longwave emissions to space (hence the greenhouse effect).

We have changed this sentence to “The atmosphere will lose more longwave radiation energy to the outer space and to the Earth’s surface as its temperature increases due to rising CO₂ and as it accumulates water vapor.” (L26)

(Please also refer to point L26, review 2)

95. See above; the global mean decomposition doesn’t actually tell us anything about how a decrease in albedo affects global evaporation. If the Bowen ratio is large at high latitudes, a change in albedo would mainly affect the sensible heat flux according to Eq. 4.

112-117: see #1 above

Please see our response to point 1

134- : see #2 above. It’s hard to interpret this slope without knowing the independent contributions from CO₂ and temperature.

Please see our response to point 2

166 - 169: The authors seem misguided here. In prescribed-SST simulations, there is no thermal coupling between the atmosphere and the ocean, so the ocean heat storage term can be quite large, both locally and globally. If SSTs are prescribed to be warmer than

their equilibrium value given the prescribed forcing, then there is a net transfer of heat from the ocean to the atmosphere (i.e., ocean heat storage is negative). This will result in a larger increase in evaporation than would realistically occur in response to greenhouse warming.

We have removed these sentences. Thank you.

Reviewer comments, second round:

Reviewer #1 (Remarks to the Author):

The authors have made numerous changes to the manuscript that have mostly addressed my previous concerns. In the revised version, more physical insight is provided and the methods are more clearly explained. I have only a few remaining minor comments, mostly related to presentation, that I recommend be addressed prior to publication of the paper.

1) While the methods are now better explained, there are a few places where additional clarification could be given:

- L395-398: It seems reasonable to assume that the albedo calculation was performed using CERES surface fluxes (as opposed to TOA fluxes, based on ref. 20), but that is not explicitly stated here. Can you clarify that here?

- It is noted that the beta (L404-406) and upward longwave (L421-423) temperature sensitivities were computed by evaluating a derivative at the observed temperature, with uncertainty corresponding to the spread in historical temperature – I feel this could use more clarification. Is the sensitivity computed for each year of GISTEMP (i.e., using each year to evaluate the derivative), and the spread corresponds to the maximum and minimum sensitivities for the entire GISTEMP period?

- L420: The additional sensitivity from increasing CO₂ is given as 0.43 +/- 0.063. In the previous draft of this paper, the central value was 0.46 (which was equivalent to 1.38/3.0, the forcing for CO₂ doubling divided by climate sensitivity). Is 0.43 a mistake? Also, how is the uncertainty of 0.063 mathematically obtained?

- L428 “confidence bound on a²” – can you explain this statistical procedure a bit further and/or provide a reference?

2) I recommend rearranging some supplementary figures/tables to improve the flow/presentation:

- First, it seems the surface energy budget diagram (currently Fig. S4) should be discussed/referenced earlier in the paper. I suggest referencing this figure from the main text (perhaps around L79-87 when introducing the surface energy budget and eq. 1), and thus switching the Fig. S3 and S4 order.

- The order at which other supplementary figures/tables are mentioned (in the main text and/or methods) is inconsistent with the order the figures/tables themselves are presented. For example, Fig. S7 (L251) is referenced before Fig. S6 (L277). Fig. S8 is first referenced (L173) before either S6 or S7 (and others). Finally, Table S4 (L408) is referenced before Table S3 (L461).

Typos/writing:

L15: I suggest changing “Assessment Report 5” to “IPCC Fifth Assessment Report” (assuming this is what you are referring to).

L26: “the outer space” -> “outer space”

L61: “confident level” -> “confidence level”

L133: “tends to give” -> “tend to give”

L293 (Fig. 1 caption): “Arrows 1-5 represents” -> “Arrows 1-5 represent”

L662 (Fig. S7 caption): "model" -> "models" ?

Fig. S8: "Silar" should be "Siler" on the figure.

Reviewer #2 (Remarks to the Author):

The revised manuscript is in very good form and the responses to my comments are very appropriate. I really appreciate that the authors chose to add a large section trying to make a bridge with the atmospheric-energy-balance constraint on precipitation change; I believe it was a very necessary addition and will be of great value to the reader.

I make a few additional suggestions with respect to the connection between the two constraints, in order to clarify the new elements of reasoning and make them more robust. I also suggest a few citations that the authors could include for completeness.

– Benjamin Fildier

1 Comments

1. **Figure 5:** It seems that the $4\times\text{CO}_2$ experiment has a systematically different behavior than other transient scenarios, and you mention it quite often already. On Figure 5c, it seems that the slope might be biased low because of the abrupt $4\times\text{CO}_2$. Should we expect a much larger $\Delta P/\Delta T$ sensitivity if we remove this – useful, but largely unrealistic – scenario?
2. **L139-141:** interesting remark about the larger predictive power of composite variables..
3. **L157-158** very nice.
4. **L160** the fact that surface solar radiation change matters, combined with an earlier statement that it comes from variations in cloud cover, is very interesting because it is apparently contradictory to what the atmospheric-energy-budget thinking suggests: that clouds don't matter for the energetic constraint on $\Delta P/\Delta T$, only the clear-sky radiative balance matters. Or maybe it is too simple a shortcut on my part? That would be worth commenting briefly.
5. **L202 “let us suppose that the net SW radiation at TOA does not change”** that is a useful approximation for making the bridge, but could you inform the magnitude of spread in net TOA SW and compare it with the magnitude of spread in net surface SW, for reference? That would be useful in order to assess the validity of L223 stating “shortwave absorption” and not “net SW flux at the surface” as a source of spread.
6. **L211 “the absorbed shortwave largely controls inter-model spread in $\Delta P/\Delta T$ in $4\times\text{CO}_2$ experiments”:**
Takahashi (2009) does not analyze $4\times\text{CO}_2$ experiments and it seems that Deangelis et al. (2015) combine it with other experiments in their analysis of spread. For the spread in the SW_{abs} component of $\Delta P/\Delta T$, you could also cite Fildier and Collins (2015).

As a general note on this point, I believe it is actually hard to distinguish what the main contributor of spread in $\Delta P/\Delta T$ is: LW_c or SW_{abs} ? Takahashi (2009) emphasizes the role of SW_{abs} and argues that LW_c is “more robustly constrained by longwave physics”, but Deangelis et al. (2015) report that “[LW_c and SW_{abs}] each account for a substantial intermodel spread in $\Delta P/\Delta T$ ”, and Pendergrass and Hartmann (2014) show that the spread in $\Delta LW_c/\Delta T$ is actually larger than that of $\Delta SW_{abs}/\Delta T$ (because it is governed by the spread across models’ lapse rate feedbacks, despite the robustness in longwave radiative transfer physics that Takahashi had highlighted). That said, the correlation you find with K_{\downarrow} does make sense.

7. **L217** that is an interesting counter-intuitive remark to make indeed. Any simple hypothesis to explain that?
8. **L223** would it make sense to make a statement linking the spread in longwave loss at the TOA to the spread in atmospheric longwave cooling, similarly to what you do with the SW component? Similarly to comment #5 could you compare the magnitudes of the spreads in net LW fluxes between the TOA and the surface? If you do, you could again cite Pendergrass and Hartmann (2014) who investigates the spread in $\Delta LW_c/\Delta T$.
9. **L281** you could also cite Bony and Dufresne (2005); Sherwood et al. (2014)

References

- Sandrine Bony and Jean Louis Dufresne. Marine boundary layer clouds at the heart of tropical cloud feedback uncertainties in climate models. *Geophysical Research Letters*, 32(20):1–4, oct 2005. ISSN 00948276. doi: 10.1029/2005GL023851. URL <https://agupubs.onlinelibrary.wiley.com/doi/full/10.1029/2005GL023851><https://agupubs.onlinelibrary.wiley.com/doi/abs/10.1029/2005GL023851><https://agupubs.onlinelibrary.wiley.com/doi/10.1029/2005GL023851>.
- Anthony M. Deangelis, Xin Qu, Mark D. Zelinka, and Alex Hall. An observational radiative constraint on hydrologic cycle intensification. *Nature*, 528(7581):249–253, dec 2015. ISSN 14764687. doi: 10.1038/nature15770. URL <http://www.nature.com/articles/nature15770>.
- B. Fildier and W. D. Collins. Origins of climate model discrepancies in atmospheric shortwave absorption and global precipitation changes. *Geophysical Research Letters*, 42(20):8749–8757, 2015. ISSN 19448007. doi: 10.1002/2015GL065931.
- Angeline G Pendergrass and Dennis L Hartmann. The Atmospheric Energy Constraint on Global-Mean Precipitation Change. *Journal of Climate*, 27(2):757–768, jan 2014. ISSN 0894-8755. doi: 10.1175/JCLI-D-13-00163.1. URL <http://journals.ametsoc.org/doi/abs/10.1175/JCLI-D-13-00163.1>.
- Steven C Sherwood, Sandrine Bony, and Jean-Louis Dufresne. Spread in model climate sensitivity traced to atmospheric convective mixing. *Nature*, 505(7481):37–42, 2014. ISSN 1476-4687. doi: 10.1038/nature12829. URL <http://www.ncbi.nlm.nih.gov/pubmed/24380952>.
- Ken Takahashi. The Global Hydrological Cycle and Atmospheric Shortwave Absorption in Climate Models under CO2 Forcing. *Journal of Climate*, 22(21):5667–5675, nov 2009. ISSN 0894-8755. doi: 10.1175/2009JCLI2674.1. URL <http://journals.ametsoc.org/doi/abs/10.1175/2009JCLI2674.1>.

Reviewer #3

I appreciate the authors' attempt to address some of my concerns. However, I still think there's a major problem with the paper that goes to the root of their claim that changes in ocean albedo have played a major role in the observed increase in ocean evaporation.

I'm afraid my previous attempt to explain the problem was not very clear, dwelling as it did on spatial variations in the bowen ratio, so I'll try another approach. The claim that albedo is important is based entirely on the global-mean Priestly-Taylor decomposition. This might make sense if the input variables in the decomposition were relatively uniform across the globe, but in fact the spatial variability is very large: the balance of terms in the arctic, for example, is much different than the balance of terms in the tropics. I worry that global averaging of the individual variables used in the decomposition gives a misleading picture of what's happening physically.

A red flag is that melting sea ice is claimed to contribute to an increase in *global* evaporation of $0.72 \text{ W/m}^2/\text{K}$. That implies enormous increases in evaporation at the sea-ice edge. I wouldn't be surprised if changes of this order have occurred in absorbed solar radiation. However, I doubt that most of the additional absorbed radiation is balanced by an increase in LH flux locally. If it's not, it raises serious questions about the value of the global decomposition.

Out of curiosity, I looked at the changes in annual-mean ocean evaporation over the last 40 years within the ERA5 model. The figure below shows the average evaporation over the decade 2009-2018 minus the average evaporation over the decade 1979-1988 (in $\text{W/m}^2/\text{K}$). According to ERA5, the large majority of the increase in evaporation over the past 4 decades has occurred outside of polar regions, with changes poleward of 60 degrees accounting for less than 3% of the global change in evaporation. I have no idea how trustworthy the ERA evaporation data is, but regardless, it highlights a crucial gap in the authors' argument: if sea ice is largely responsible for the observed increase in global evaporation, we should see very large increases in evaporation locally where the sea ice has retreated. I think the authors need to demonstrate that if they can.

If they can't, a possible way forward might be to perform the decomposition locally and then take the global mean of each term to compute the global-mean contribution. The narrative will likely change, but it could be valuable.

I won't review the rest of the paper until this major concern is addressed, but there are a couple other points I'd like to reiterate from my earlier review. The first relates to a sentence on line 26: the atmosphere does not radiate more energy to space as it accumulates CO₂—it radiates less! That's the reason for greenhouse warming: the earth is absorbing more energy than it is emitting. As the climate warms, longwave emissions increase until equilibrium is restored. Perhaps the authors know this but the sentence as written is misleading. Second, because the heat capacity of the atmosphere is quite small compared with that of the ocean, the radiation imbalance at TOA must be quite close to ocean heat uptake in the global mean. From the perspective of the surface energy budget, it's the ocean heat uptake that drives the decrease in global precipitation as a direct result of CO₂ forcing.

Response to reviews of NCOMMS-20-19680 “Constraining the sensitivity of global precipitation to warming with ocean surface energy balance”

(Note: The review comments are in bold font, our responses are in regular font, and changes made to the manuscript are in quotations and highlighted in blue.)

Review 1

The authors have made numerous changes to the manuscript that have mostly addressed my previous concerns. In the revised version, more physical insight is provided and the methods are more clearly explained. I have only a few remaining minor comments, mostly related to presentation, that I recommend be addressed prior to publication of the paper.

Thank you.

1) While the methods are now better explained, there are a few places where additional clarification could be given:

- L395-398: It seems reasonable to assume that the albedo calculation was performed using CERES surface fluxes (as opposed to TOA fluxes, based on ref. 20), but that is not explicitly stated here. Can you clarify that here?

We have now explicitly stated this in the revision. (L396)

- It is noted that the beta (L404-406) and upward longwave (L421-423) temperature sensitivities were computed by evaluating a derivative at the observed temperature, with uncertainty corresponding to the spread in historical temperature – I feel this could use more clarification. Is the sensitivity computed for each year of GISTEMP (i.e., using each year to evaluate the derivative), and the spread corresponds to the maximum and minimum sensitivities for the entire GISTEMP period?

We have changed the text to:

“The β temperature sensitivity was obtained from the derivative of this function with respect to observed global mean T . It was computed for each year, and its spread corresponds to one standard deviation of the interannual variability.” (L405)

“The L_1 temperature sensitivity was given by the derivative of the Stefan-Boltzmann Law. The calculation was done annually using observed global mean temperature. Its uncertainty corresponds to one standard deviation of the interannual variability.” (L421)

- L420: The additional sensitivity from increasing CO₂ is given as 0.43 +/- 0.063. In the previous draft of this paper, the central value was 0.46 (which was equivalent to 1.38/3.0, the forcing for CO₂ doubling divided by climate sensitivity). Is 0.43 a mistake? Also, how is the uncertainty of 0.063 mathematically obtained?

The correct number is 0.43 (= 1.38 / 3.2). The uncertainty (one standard deviation) was calculated by dividing the 90% bound -- provided by IPCC AR5 (Table 9.5 on page 818) – with 1.64. We made a small mistake in this calculation. The corrected uncertainty is $\pm 0.11 \text{ W m}^{-2} \text{ K}^{-1}$. (L419)

- L428 “confidence bound on a₂” – can you explain this statistical procedure a bit further and/or provide a reference?

We have expanded the explanation to

“The first derivative of OHC with respect to t gives heat storage change or total heat flux (in W) into the water column, and the second derivative (or a_2) represents the time rate of change of this total heat flux. Dividing the coefficient of the quadratic term a_2 by the ocean area gives the time rate of change of the heat flux into the water column per unit surface area (in $\text{W m}^{-2} \text{ s}^{-1}$), and by multiplying this rate by the length of the observational period (1955 to 2017), we obtained ΔG .” (L425).

Figure R1 below shows the actual regression results. The ΔG presented in the paper is the average of these two datasets

Figure R1: Quadratic fit of the ISH (Ishii et al., 2017, Sci. Online Lett. Atmos. 13, 163) and CHG (Cheng et al., 2017, Sci. Adv. 3, e1601545) ocean heat content datasets.

2) I recommend rearranging some supplementary figures/tables to improve the flow/presentation:

- First, it seems the surface energy budget diagram (currently Fig. S4) should be discussed/referenced earlier in the paper. I suggest referencing this figure from the main text (perhaps around L79-87 when introducing the surface energy budget and eq. 1), and thus switching the Fig. S3 and S4 order.

Done.

- The order at which other supplementary figures/tables are mentioned (in the main text and/or methods) is inconsistent with the order the figures/tables themselves are presented. For example, Fig. S7 (L251) is referenced before Fig. S6 (L277). Fig. S8 is first referenced (L173) before either S6 or S7 (and others). Finally, Table S4 (L408) is referenced before Table S3 (L461).

We have rearranged figure and table sequences according to the order of presentation.

Typos/writing:

L15: I suggest changing “Assessment Report 5” to “IPCC Fifth Assessment Report” (assuming this is what you are referring to).

L26: “the outer space” -> “outer space”

L61: “confident level” -> “confidence level”

L133: “tends to give” -> “tend to give”

L293 (Fig. 1 caption): “Arrows 1-5 represents” -> “Arrows 1-5 represent”

L662 (Fig. S7 caption): “model” -> “models” ?

Fig. S8: “Silar” should be “Siler” on the figure.

Done. Thank you for your careful reading of our submission.

Review 2

The revised manuscript is in very good form and the responses to my comments are very appropriate. I really appreciate that the authors chose to add a large section trying to make a bridge with the atmospheric-energy-balance constraint on precipitation change; I believe it was a very necessary addition and will be of great value to the reader.

I make a few additional suggestions with respect to the connection between the two constraints, in order to clarify the new elements of reasoning and make them more robust. I also suggest a few citations that the authors could include for completeness.

Thank you.

1 Comments

1. Figure 5: It seems that the 4×CO₂ experiment has a systematically different behavior than other transient scenarios, and you mention it quite often already. On Figure 5c, it seems that the slope might be biased low because of the abrupt 4×CO₂. Should we expect a much larger $\Delta P/\Delta T$ sensitivity if we remove this - useful, but largely unrealistic scenario?

You are correct. The regression slope changes from 0.982 to 1.290 if the 4×CO₂ results are excluded. This point is now noted in the figure caption.

2. L139-141: interesting remark about the larger predictive power of composite variables.

3. L157-158 very nice.

Thank you.

4. L160 the fact that surface solar radiation change matters, combined with an earlier statement that it comes from variations in cloud cover, is very interesting because it is apparently contradictory to what the atmospheric-energy-budget thinking suggests: that clouds don't matter for the energetic constraint on $\Delta P/\Delta T$, only the clear-sky radiative balance matters. Or maybe it is too simple a shortcut on my part? That would be worth commenting briefly.

Thank you for this interesting observation. Previous studies based on the atmospheric energy constraint indeed show the importance of clear-sky radiative balance in controlling the global P temperature sensitivity. Those studies are mostly restricted to examining inter-model spread in a specific scenario, such as AR4 A1b (Pendergrass & Hartmann, GRL 39, L01703), 4 x CO₂ (DeAngelis et al., ref 6) and 1pctCO₂ (Pendergrass and Hartmann, ref 10). Our presentation here

focuses on inter-scenario variations. We are not aware of a similar inter-scenario examination of clear-sky atmospheric energy balance.

5. L202 “let us suppose that the net SW radiation at TOA does not change” that is a useful approximation for making the bridge, but could you inform the magnitude of spread in net TOA SW and compare it with the magnitude of spread in net surface SW, for reference? That would be useful in order to assess the validity of L223 stating “shortwave absorption” and not “net SW flux at the surface” as a source of spread.

Our original statement is not accurate because the net SW radiation at TOA does change over time. However, we find that the change in atmospheric SW absorption is correlated with the change in SW at the surface (Figure R2). Much of the discussion that follows still stands without this simplifying assumption. In the revision, we have removed this statement and have modified the wording of the rest of this paragraph accordingly. For example, the last sentence is changed to:

“Since ΔK_{\downarrow} is approximately equal to the change in atmospheric absorption of shortwave minus the change in the TOA net shortwave radiation, and ΔG is an approximation of the change in the total net radiation at the TOA, a physical interpretation of the emergent relationship in Figure 4a is that shortwave absorption (a known source of model spread⁶) and longwave loss at the TOA¹⁰ dominate the modeled P change.” (L212)

Figure R2. Comparison of changes in atmospheric absorption of SW radiation ($\Delta K_{ab} - ATM$) and in incoming SW radiation at the ocean surface (ΔK_{\downarrow}) across scenarios. Error bars are one standard deviation.

6. L211 “the absorbed shortwave largely controls inter-model spread in DP/DT in $4\times CO_2$ experiments”: Takahashi (2009) does not analyze $4\times CO_2$ experiments and it seems that Deangelis et al. (2015) combine it with other experiments in their analysis of

spread. For the spread in the SW_{abs} component of DP/DT , you could also cite Fildier and Collins (2015).

You are correct. Takahashi (2009) analyzed the $2\times\text{CO}_2$ experiment from CMIP3, not the $4\times\text{CO}_2$ experiment from CMIP5. The focus of DeAngelis et al. (2015) is actually the $4\times\text{CO}_2$ model spread (e.g. their Figure 2). This sentence is modified to:

“The finding that the absorbed shortwave largely controls inter-model spread in $\Delta P/\Delta T$ in abrupt CO_2 ($4\times\text{CO}_2$ and $2\times\text{CO}_2$) scenarios^{6,26} is supported by the $\Delta P/\Delta T$ correlation with K_{\downarrow} change (Figure 3g).” (L202)

Fildier and Collins (2015) is now cited (ref⁵⁴) on L216.

As a general note on this point, I believe it is actually hard to distinguish what the main contributor of spread in DP/DT is: LW_c or SW_{abs} ? Takahashi (2009) emphasizes the role of SW_{abs} and argues that LW_c is “more robustly constrained by longwave physics”, but Deangelis et al. (2015) report that “[LW_c and SW_{abs}] each account for a substantial intermodel spread in DP/DT ”, and Pendergrass and Hartmann (2014) show that the spread in $DLW_c=DT$ is actually larger than that of $DSW_{\text{abs}}=DT$ (because it is governed by the spread across models' lapse rate feedbacks, despite the robustness in longwave radiative transfer physics that Takahashi had highlighted). That said, the correlation you find with K_{\downarrow} does make sense.

Thank you for this insightful comment.

7. L217 that is an interesting counter-intuitive remark to make indeed. Any simple hypothesis to explain that?

We have modified this sentence to:

“The importance of atmospheric longwave cooling documented for a future transient climate¹⁰ and for the historical climate⁴⁴ is manifested in the correlation with changes in G (Figure 5c) and L_{\downarrow} (Figure 3h) because longwave loss to outer space is a large contributor to the TOA energy imbalance (and hence to G).” (L206)

8. L223 would it make sense to make a statement linking the spread in longwave loss at the TOA to the spread in atmospheric longwave cooling, similarly to what you do with the SW component? Similarly to comment #5 could you compare the magnitudes of the spreads in net LW fluxes between the TOA and the surface? If you do, you could again cite Pendergrass and Hartmann (2014) who investigates the spread in DLW_c/DT .

The spread of ΔL_{net} at the ocean surface is 2.00 W m^{-2} (1 S.D. across all CMIP5 model simulations), which is similar to that for ΔL_{\uparrow} at the TOA (1.99 W m^{-2}).

We have cited Pendergrass and Hartmann (2014) again here (L216).

9. L281 you could also cite Bony and Dufresne (2005); Sherwood et al. (2014)

Thank you. The two papers are cited (L274).

References

**Sandrine Bony and Jean Louis Dufresne. Marine boundary layer clouds at the heart of tropical cloud feedback uncertainties in climate models. *Geophysical Research Letters*, 32(20):104, oct 2005. ISSN 00948276. doi: 10.1029/2005GL023851.
URL <https://agupubs.onlinelibrary.wiley.com/doi/full/10.1029/2005GL023851>
<https://agupubs.onlinelibrary.wiley.com/doi/abs/10.1029/2005GL023851>
<https://agupubs.onlinelibrary.wiley.com/doi/10.1029/2005GL023851>.**

**Anthony M. Deangelis, Xin Qu, Mark D. Zelinka, and Alex Hall. An observational radiative constraint on hydrologic cycle intensification. *Nature*, 528(7581):249{253, dec 2015. ISSN 14764687. doi: 10.1038/nature15770.
URL <http://www.nature.com/articles/nature15770>.**

B. Fildier and W. D. Collins. Origins of climate model discrepancies in atmospheric shortwave absorption and global precipitation changes. *Geophysical Research Letters*, 42(20):8749{8757, 2015. ISSN 19448007. doi: 10.1002/2015GL065931.

**Angeline G Pendergrass and Dennis L Hartmann. The Atmospheric Energy Constraint on Global-Mean Precipitation Change. *Journal of Climate*, 27(2):757-768, jan 2014. ISSN 0894-8755. doi: 10.1175/JCLI-D-13-00163.1.
URL <http://journals.ametsoc.org/doi/abs/10.1175/JCLI-D-13-00163.1>.**

**Steven C Sherwood, Sandrine Bony, and Jean-Louis Dufresne. Spread in model climate sensitivity traced to atmospheric convective mixing. *Nature*, 505(7481):37-42, 2014. ISSN 1476-4687. doi: 10.1038/nature12829.
URL <http://www.ncbi.nlm.nih.gov/pubmed/24380952>.**

**Ken Takahashi. The Global Hydrological Cycle and Atmospheric Shortwave Absorption in Climate Models under CO₂ Forcing. *Journal of Climate*, 22(21):5667-5675, nov 2009. ISSN 0894-8755. doi: 10.1175/2009JCLI2674.1.
URL <http://journals.ametsoc.org/doi/abs/10.1175/2009JCLI2674.1>.**

Review 3

I appreciate the authors' attempt to address some of my concerns. However, I still think there's a major problem with the paper that goes to the root of their claim that changes in ocean albedo have played a major role in the observed increase in ocean evaporation.

I'm afraid my previous attempt to explain the problem was not very clear, dwelling as it did on spatial variations in the bowen ratio, so I'll try another approach. The claim that albedo is important is based entirely on the global-mean Priestly-Taylor decomposition. This might make sense if the input variables in the decomposition were relatively uniform across the globe, but in fact the spatial variability is very large: the balance of terms in the arctic, for example, is much different than the balance of terms in the tropics. I worry that global averaging of the individual variables used in the decomposition gives a misleading picture of what's happening physically.

A red flag is that melting sea ice is claimed to contribute to an increase in *global* evaporation of 0.72 W/m²/K. That implies enormous increases in evaporation at the sea-ice edge. I wouldn't be surprised if changes of this order have occurred in absorbed solar radiation. However, I doubt that most of the additional absorbed radiation is balanced by an increase in LH flux locally. If it's not, it raises serious questions about the value of the global decomposition.

Out of curiosity, I looked at the changes in annual-mean ocean evaporation over the last 40 years within the ERA5 model. The figure below shows the average evaporation over the decade 2009-2018 minus the average evaporation over the decade 1979-1988 (in W/m²/K). According to ERA5, the large majority of the increase in evaporation over the past 4 decades has occurred outside of polar regions, with changes poleward of 60 degrees accounting for less than 3% of the global change in evaporation. I have no idea how trustworthy the ERA evaporation data is, but regardless, it highlights a crucial gap in the authors' argument: if sea ice is largely responsible for the observed increase in global evaporation, we should see very large increases in evaporation locally where the sea ice has retreated. I think the authors need to demonstrate that if they can.

If they can't, a possible way forward might be to perform the decomposition locally and then take the global mean of each term to compute the global-mean contribution. The narrative will likely change, but it could be valuable.

Priestly-Taylor model

Thank you for these constructive comments. We would like to begin our response by clarifying one minor point about the PT model. This model (after modification according to the Objectively Analyzed Air-sea Flux dataset by Yang & Roderick, ref 12) is only used to determine the Bowen ratio temperature sensitivity. The actual decomposition is made with the energy balance equation (Equations 4 & 5).

Errors due to spatial averaging

To reiterate some of the points we made in the first round of review response, the reason for why we use the global framework is that at the global scale, the surface energy balance provides a complete constraint on P without the need to consider motion dynamics. In this regard, the framework is similar to the global atmospheric energy balance used by other researchers to constrain global P. In contrast, a regional or local analysis would be confounded by dynamic processes for which observational data are either lacking or much more uncertain (more on this point later).

The concern about errors due to spatial averaging is a valid one. These errors appear small for the global $\Delta P/\Delta T$ because excellent agreement is achieved between the offline diagnostic calculations and direct model outputs (Figure 2). Our offline method has also successfully reproduced the $\Delta P/\Delta T$ reported by Fläschner (ref⁵) for the $4\times\text{CO}_2$ scenario from the atmospheric energy balance constraint, further supporting the robustness of the calculated $\Delta P/\Delta T$. But we cannot rule out errors in and error compensation among individual component contributions to

the global P change; this issue is discussed extensively (e.g., L122, L129, L220, L442, L491, L497).

Evaporation at sea-ice edges

We agree that an inference from our results is the hypothesis that large evaporation rate occurs at sea-ice edges. A comprehensive test of this hypothesis is, however, beyond the scope of this study, in part because ocean evaporation from reanalysis data products is unconstrained by energy balance. We are encouraged by the evidence, albeit indirect, for this hypothesis from several terrestrial studies. In an eddy-covariance experiment at the Great Lakes, lake E in a winter with low ice coverage was substantially higher than in a winter with normal ice coverage, resulting in a large overall increase in annual E of 8 W m^{-2} or 16% (Blanken et al., J Great Lakes Research, 37: 707). In an energy diagnostic analysis by Wang et al. (2018, Nature Geoscience, 11: 410), albedo reduction via ice shrinkage contributes to the rapid increase in lake E due to rising temperatures. Interestingly, the contribution of albedo change to the global lake E temperature sensitivity ($0.75 \text{ W m}^{-2} \text{ K}^{-1}$) is comparable to the value reported here.

Another line of indirect evidence in support of the sea-ice edge hypothesis is provided by the historical CMIP modeling results. Large ensemble mean albedo change is found at latitudes north 60° N and south of 60° S (Figure R3, left-most panel). At these latitudes, the ensemble mean E change is consistently positive and large (Figure R3, right-most panel).

Figure R3: Latitudinal patterns of changes (mean of last 10 y minus mean of first 10 y) in surface energy balance variables according to CMIP5 historical simulations. Shaded boundary indicates inter-model spread (± 1 standard deviation).

Local/regional decomposition

Thank you for suggesting a local/regional decomposition analysis. Unfortunately, this cannot be done with the existing data products. At local and regional scales, the energy balance equation takes the same form as at the global scale:

$$R_n = (1 - a)K_{\downarrow} + L_{\downarrow} - L_{\uparrow} = H + E_0 + G$$

(Equation 5). The heat flux from the atmosphere to the water column G now consists of two terms, as

$$G = T + S$$

where T is lateral heat transport by ocean currents (which is dominant) and S is local change in ocean heat content (which is minor; Trenberth et al., 2019, J Climate, 32: 4567). No gridded observational data exist on T . Furthermore, observational data on S are highly uncertain for polar regions (von Schuckmann et al. 2020, ref 42).

Local/regional decomposition can be done with CMIP5 modeling outputs. (In CMIP5 models, the gridded G is the sum of T and S .) Here, in response to your comment we have conducted a regional analysis with CMIP historical simulations. In this analysis, the ocean grids are divided into two groups: those belonging to mid- and low-latitude regions (between 60° S and 60° N) and those belonging to high-latitude regions (north of 60° N and south of 60° S). We choose 60° S and 60° N as the boundaries because albedo change occurs mostly at latitudes north 60° N and south of 60° S (Figure R3). The decomposition is performed for each group and the result is weighted by the area fraction of each to obtain a global mean value (Figures R4 & R5). The albedo contribution from the two-region analysis is smaller than that from the global analysis. The reduction in the albedo component is offset by less negative contributions from changes in shortwave radiation and in ocean heat storage. The total $\Delta P/\Delta T$ is unaffected, as the overall $\Delta P/\Delta T$ from the two-region analysis ($0.51 \pm 0.76 \text{ W m}^{-2} \text{ K}^{-1}$; Figure R4, panel b) is nearly identical to that from the global analysis ($0.52 \pm 0.81 \text{ W m}^{-2} \text{ K}^{-1}$; Figure R4, panel a).

Figure R4 (New Figure S9): Comparison of regional and global analysis using CMIP5 historical simulations. a, Component contributions to global precipitation temperature sensitivity calculated with Equation (1) using global

mean values as inputs. b, Component contributions from a regional diagnostic analysis, where Equation (1) was applied separately to polar (north 60° N and south of 60° S) and non-polar grids (between 60° N and of 60° S), and the result was weighted by the area fraction of each group to give the global mean value.

Figure R5: Comparison of $\Delta P/\Delta T$ from global (x-axis) and that from the two-region diagnostic analysis for CMIP historical climate. Each data point represents one CMIP model.

ERA5

We appreciate your drawing our attention to the ERA5 evaporation data. However, we respectfully argue against using the data to diagnose local evaporation changes. In agreement with your data map, we find that in ERA5, the largest zonal mean ocean E change (mean of 2010-2019 minus mean of 1980-1989) occurs at low latitudes (Figure R6, right-most panel). However, the E changes are not correlated to changes in the key energy balance variables (albedo, net shortwave radiation, net longwave radiation); instead they are almost all explained by local changes in the energy balance residual (Figure R6, second panel from right), computed here as $\text{Res} = \text{Rn} - \text{H} - \text{E}$, where Rn is net radiation at the surface, and H and E are surface sensible and latent heat fluxes, respectively. Like other reanalysis data products, ERA5 ignores the surface energy balance constraint on E for ocean tiles. Instead, it computes E (and H) from specified sea surface temperature with a bulk aerodynamic transfer method. The ERA5 global ocean mean residual is 8.4 W m^{-2} (mean for 1989 to 2008), which is an order of magnitude too large in comparison with the observed ocean heat content change. Because of the lack of energy balance constraint, the energy diagnostic analysis would result in misleading interpretation.

Figure R6: Latitudinal patterns of changes (mean of 2010 to 2019 minus mean of 1980 to 1989) in surface energy balance variables according to ERA5.

Component contributions versus total P sensitivity

We regret that our writing has caused a number of misinterpretations, including (1) that sea ice change is largely responsible for the global E change, and (2) that most of the additional absorbed radiation is balanced by an increase in latent heat flux locally. We put too much emphasis on the contribution of albedo change, but in fact the largest positive contribution to global P change actually comes from change in longwave radiation (Figure 2a). Furthermore, the albedo contribution is less than half in magnitude of the solar radiation component. The total change in global E (and P) is a net balance of these positive and negative components, so we agree that it was misleading to draw attention to one component but overlook the others. In this revision, we have rewritten the text about the role of sea ice/albedo change to avoid confusion.

Because we use a global framework, we cannot say much about local energy balance. That Bowen ratio is reasonably large (0.70) at latitudes north of 60° N and south of 60° S (**L266**) implies that about 40% of the additional shortwave (and longwave) radiation energy in the polar regions is dissipated to the atmosphere as sensible heat.

Perhaps it is helpful here to reiterate the main goal of this study, which is to provide an energetic constraint on the historical global P change. Although the component contributions to the historical $\Delta P/\Delta T$ may be subject to multiple interpretations, our conclusion about the sign and magnitude of $\Delta P/\Delta T$ (**L8**) is robust. There are two reasons for this. First, the emergent relationship (Figure 4a) yields nearly the same $\Delta P/\Delta T$ as the energy decomposition analysis (**L190**). Most notably, this relationship does not involve the (disputed) albedo term. Second, $\Delta P/\Delta T$ calculated from global decomposition agrees with that from regional decomposition (Figure R4; thank you once again for this suggestion).

Revision summary

Regarding the role of albedo/sea ice:

- Original: “The reduction in ocean surface albedo associated with melting of the sea ice is a large contributor to the P temperature sensitivity $\Delta P/\Delta T$; without it, the historical $\Delta P/\Delta T$ would be negative.” Changed to: “The reduction in ocean surface albedo associated with melting of sea ice is a positive contributor to the P temperature sensitivity $\Delta P/\Delta T$.” (L10)
- Original: “Ocean albedo change plays a large role, contributing $0.72 \text{ W m}^{-2} \text{ K}^{-1}$ to the overall sensitivity (Figure 2a); without this contribution, $\Delta P/\Delta T$ would be negative. Melting of the sea ice has long been recognized as a positive feedback that amplifies warming. Our result shows that the same process also increases the global precipitation temperature sensitivity.” Changed to: “Ocean albedo change contributes positively to the overall sensitivity (Figure 2a). Melting of the sea ice has long been recognized as a positive feedback that amplifies warming. Our result suggests that the same process may also increase the global precipitation temperature sensitivity.” (L103)
- Deleted: “As with the observation-based analysis, the ocean albedo reduction plays a large role. Previously, kernel decomposition of the atmospheric energy balance suggests a negligible role of surface albedo^{5, 21}, perhaps because the ocean albedo signal is hidden in the large residual of the decomposition or because its role is masked by land albedo changes.” (Original L123)

Spatial averaging errors

- Revised as: “The consistency between the online and offline calculations indicate that Equation (1) is a robust decomposition procedure and that errors in the global $\Delta P/\Delta T$ arising from spatial averaging of input variables may be small. The offline $\Delta P/\Delta T$ from the surface energy balance (Equation 1) for the $4\times\text{CO}_2$ scenario ($2.18 \pm 0.21 \text{ W m}^{-2} \text{ K}^{-1}$; Figure 2g) agrees well with the $\Delta P/\Delta T$ diagnosed from the atmospheric energy balance ($2.03 \text{ W m}^{-2} \text{ K}^{-1}$; ref⁵), offering further support for the surface diagnostic method.” (L489)

Regional decomposition analysis

- Added: “To further investigate possible errors due to spatial averaging, we performed a regional diagnostic analysis using CMIP historical simulations. At regional and local scales, the heat flux from the atmosphere to the water column G consists of lateral heat transport by ocean currents and time change in local ocean heat content⁵³. Regional analysis is not feasible with observational data because no gridded data exist on the transport term, but it can be done with CMIP modeling outputs as the modeled G includes both lateral heat transport and local heat storage. In this analysis, the ocean grids were divided into two groups: those belonging to mid- and low-latitude regions (between 60° S and 60° N) and those belonging to high-latitude regions (north 60° N and south of 60° S).

The decomposition was performed for each group and the result was weighted by the area fraction of each to obtain a global mean value (Figure S9). The albedo contribution from the two-region analysis is smaller than that from the global analysis. The reduction in the albedo component is offset by less negative contributions from changes in shortwave radiation and in ocean heat storage. The total $\Delta P/\Delta T$ is unaffected, as $\Delta P/\Delta T$ from the two-region analysis ($0.51 \pm 0.76 \text{ W m}^{-2} \text{ K}^{-1}$; Figure S9 panel b) is nearly identical to that from the global analysis ($0.52 \pm 0.81 \text{ W m}^{-2} \text{ K}^{-1}$; Figure S9 panel a).” (L497)

I won't review the rest of the paper until this major concern is addressed, but there are a couple other points I'd like to reiterate from my earlier review. The first relates to a sentence on line 26: the atmosphere does not radiate more energy to space as it accumulates CO₂—it radiates less! That's the reason for greenhouse warming: the earth is absorbing more energy than it is emitting. As the climate warms, longwave emissions increase until equilibrium is restored. Perhaps the authors know this but the sentence as written is misleading. Second, because the heat capacity of the atmosphere is quite small compared with that of the ocean, the radiation imbalance at TOA must be quite close to ocean heat uptake in the global mean. From the perspective of the surface energy budget, it's the ocean heat uptake that drives the decrease in global precipitation as a direct result of CO₂ forcing.

Longwave cooling:

We have changed this sentence to (L26): “The atmosphere will lose more longwave radiation energy to the Earth's surface as its temperature increases due to rising CO₂ and as it accumulates water vapor^{9,10}.” The papers by Stephens & Hu (2010) and Pendergrass & Hartmann (2014) are cited here as supporting references (refs 9 and 10).

The reviewer is correct that if the change is referenced to the state before a CO₂ perturbation, the atmosphere will radiate less longwave energy to outer space after the perturbation. What we had in mind as the reference state was the first 10 years after the CO₂ perturbation, and the change in longwave emissions at the TOA (ΔL_{\uparrow})_{TOA} was the difference between a future state and this reference time frame. According to CMIP5 simulations, because higher atmospheric temperature in the last 10-years of model simulations, (ΔL_{\uparrow})_{TOA} is generally positive (Figure R7). However, such nuances were lost in our attempt to keep the writing concise.

Figure R7: Change (mean of last 10 y minus mean of first 10 y) in the outgoing longwave radiation at the TOA (panel a) and net longwave radiation at the ocean surface (panel b). Error bars are ± 1 standard deviation.

TOA energy imbalance

Our results are broadly consistent with this interpretation:

- “On the other hand, the observed increase in ocean heat storage weakens the historical P .” (L11)
- Ocean heat uptake makes a negative contribution to the global $\Delta P/\Delta T$ under historical, RCP4.5, RCP6.0 and RCP8.5 scenarios (Figure 2).
- “The sudden quadrupling of atmospheric CO₂ causes a large radiation imbalance at the top of the atmosphere and a similarly large heat flux into the ocean (multi-model mean $G = 6.52 \text{ W m}^{-2}$ in the first 10 simulation years)...” (L164)
- “...the TOA radiation imbalance can be approximated by the ocean heat storage G because G explains $\sim 90\%$ of the imbalance historically⁴¹ and more in the future⁴² ...” (L198)

Reviewer comments, third round:

Reviewer #1 (Remarks to the Author):

The authors have further improved the manuscript, particularly with regard to the physical interpretation of the results and clarity of the methods. Apart from one minor typo (below), I have no further comments.

In the Table S2 caption (L732): I think this should be a reference to Fig. S3, not S4.

Reviewer #2 (Remarks to the Author):

I thank the authors for accounting for all my suggestions and incorporating suggested edits in the new version of the manuscript. As far as I am concerned, I recommend that the article be published once the comments from other reviewers have been addressed.

Reviewer #3 (Remarks to the Author):

I have read the entire manuscript and the authors' response to all reviewer comments. I appreciate very much the authors' thorough response to my comments and the revisions they made to address them. In particular, I think the regional decomposition in CMIP5 is a very nice addition.

I think the manuscript is in good shape, and I would support publication at this stage.

Response to reviews of NCOMMS-20-19680B “Constraining the sensitivity of global precipitation to warming with ocean surface energy balance”

(Note: The review comments are in bold font, our responses are in regular font.)

Reviewer #1

The authors have further improved the manuscript, particularly with regard to the physical interpretation of the results and clarity of the methods. Apart from one minor typo (below), I have no further comments.

In the Table S2 caption (L732): I think this should be a reference to Fig. S3, not S4.

Done.

Reviewer #2

I thank the authors for accounting for all my suggestions and incorporating suggested edits in the new version of the manuscript. As far as I am concerned, I recommend that the article be published once the comments from other reviewers have been addressed.

Thank you.

Reviewer #3

I have read the entire manuscript and the authors' response to all reviewer comments. I appreciate very much the authors' thorough response to my comments and the revisions they made to address them. In particular, I think the regional decomposition in CMIP5 is a very nice addition.

I think the manuscript is in good shape, and I would support publication at this stage.

Thank you.